# Unveiling the dynamics of little-bang nucleosynthesis

Kai-Jia Sun [1,2] ✉, Rui Wang[1,3], Che Ming Ko [4] ✉ & Yu-Gang Ma [1,2] ✉ & Chun Shen[5,6]

High-energy nuclear collisions provide a unique site for the synthesis of both nuclei and antinuclei at temperatures of $kT \approx 100-150$ MeV. In these little bangs of transient collisions, a quark-gluon plasma (QGP) of nearly vanishing viscosity is created, which is believed to have existed in the early universe within the first few microseconds after the Big Bang. Analyses of identified particles produced in these little bangs based on the statistical hadronization model for the QGP have suggested that light (anti)nuclei are produced from the QGP as other hadrons and their abundances are little affected by later hadronic dynamics. Here, we find a strong reduction of the triton yield by about a factor of 1.8 in high-energy heavy-ion collisions based on a kinetic approach that includes the effects of hadronic re-scatterings, particularly that due to pion-catalyzed multi-body reactions. This finding is supported by the latest experimental measurements and thus unveils the important role of hadronic dynamics in the little-bang nucleosynthesis.

Light nuclei such as deuteron ($d$) and helium ($^3$He and $^4$He) in the universe were largely formed in the big-bang nucleosynthesis during the early universe through a sequence of two-body reactions like $pn \leftrightarrow \gamma d$ at temperatures less than 1 MeV[1] in the convention of $\hbar = k = c = 1$. On the other hand, high-energy nuclear collisions, in which both matter and antimatter can be produced, provide a unique site for the synthesis of nuclei and antinuclei at temperatures reaching the order of $T$-100 MeV[2]. This 'little-bang' nucleosynthesis has recently attracted much attention because of its relevance to the test of matter-antimatter asymmetry[3,4], the indirect searches of dark matter[5], etc.

Figure 1 sketches the time evolution of the bulk matter in high-energy nuclear collisions. After a brief pre-equilibrium stage of about 1 fm/$c$, the system equilibrates into a fluid-like medium of deconfined quarks and gluons named the quark-gluon plasma (QGP), which is believed to have existed in the early universe a few microseconds after the Big Bang. The QGP then expands hydrodynamically and makes a transition to a gas of hadrons, which is further followed by their re-scatterings and resonance decays. Quantum chromodynamics (QCD),

which is the theory for the strong interaction in nature, predicts that the transition from a QGP of vanishing baryon chemical potential, corresponding to that produced in ultra-relativistic heavy-ion collisions, to a hadron gas is a smooth crossover characterized by $T = 156.5 \pm 1.5$ MeV[6].

A well-established framework for describing particle production in these little bangs is the statistical hadronization of the created QGP, which involves only a few parameters like its volume, temperature, and baryon chemical potential[7]. Analyses of experimental data based on this statistical hadronization model (SHM) have shown that light (anti)nuclei and other stable hadrons share a common chemical freeze-out temperature that coincides with the QCD transition temperature and thus exceeds the light nuclei binding energies by orders of magnitude[8]. This interesting result has led to the conclusion that light nuclei are produced from the QGP at its hadronization and the post-hadronization dynamics has only small effects on their final abundance. This result is supported by studies using the Saha equation[9] and the rate equation[10] as well as the more microscopic transport approach[11].

[1]Key Laboratory of Nuclear Physics and Ion-beam Application (MOE), Institute of Modern Physics, Fudan University, Shanghai 200433, China. [2]Shanghai Research Center for Theoretical Nuclear Physics, NSFC and Fudan University, Shanghai 200438, China. [3]Shanghai Institute of Applied Physics, Chinese Academy of Sciences, Shanghai 201800, China. [4]Cyclotron Institute and Department of Physics and Astronomy, Texas A&M University, College Station, TX 77843, USA. [5]Department of Physics and Astronomy, Wayne State University, Detroit, MI 48201, USA. [6]RIKEN BNL Research Center, Brookhaven National Laboratory, Upton, NY 11973, USA. ✉e-mail: kjsun@fudan.edu.cn; mayugang@fudan.edu.cn

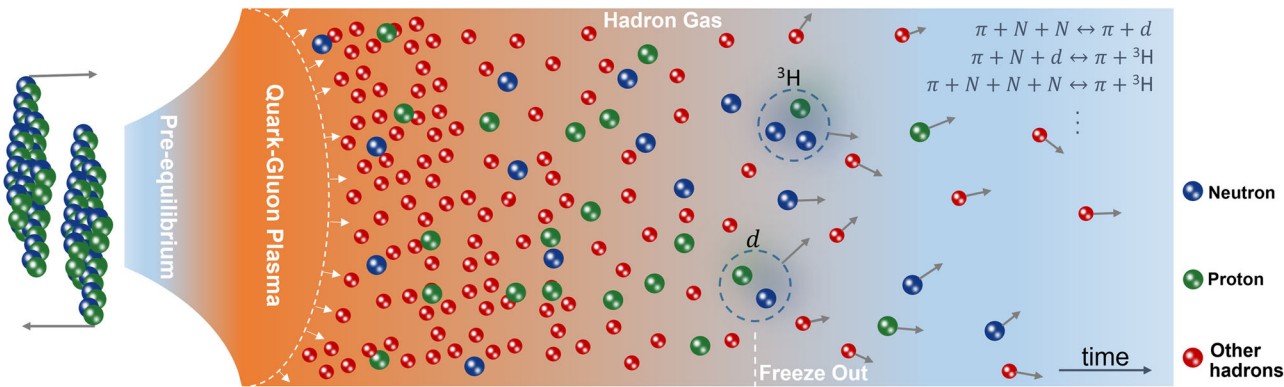

**Fig. 1 | Time evolution of high-energy nuclear collisions.** Shown from left to right are the different stages undergone by the produced hot dense matter, i.e., the pre-equilibrium dynamics, quark-gluon plasma expansion, hadronization of the quark-gluon plasma (dash line), and subsequent hadronic re-scatterings and resonance decays including the pion-catalyzed reactions for nucleosynthesis. Gray arrows denote the velocity directions of freeze-out particles.

In this article, we report, however, a surprising finding that hadronic re-scatterings can lead to a strong reduction of the triton ($^3$H) yield from its thermally equilibrated value predicted by the SHM. We describe the dynamics of light nuclei production by extending the relativistic kinetic equations (RKE)[12] in the real-time many-body Green's-function formalism to bound states[13–15] by including their dissociation and regeneration processes in the collision integrals. In Au + Au collisions at center-of-mass energies per nucleon pair of $\sqrt{s_{NN}} = 7.7 - 200$ GeV as well as in Pb + Pb collisions at even higher collision energies, the dominant hadronic re-scattering processes for deuteron and triton production are the pion-catalyzed reactions like $\pi NN \leftrightarrow \pi d$, $\pi Nd \leftrightarrow \pi^3$H, and the more complicated $\pi NNN \leftrightarrow \pi^3$H. With collision integrals of these multi-particle reactions evaluated using a stochastic method, we can obtain the full dynamics of little-bang nucleosynthesis. We find that the effect of hadronic re-scatterings on deuteron production is small, confirming the results from previous studies[9–11]. However, the initial triton number produced from the QGP is reduced by about a factor of 1.8 during the hadronic matter expansion, leading to a result that is in excellent agreement with the latest measurements by the STAR Collaboration at the Relativistic Heavy Ion Collider (RHIC)[16] and the ALICE Collaboration at the Large Hadron Collider (LHC)[17].

## Results

### Hadronic effects on deuteron and triton production in Au + Au collisions

We first consider light nuclei production in Au + Au collisions at $\sqrt{s_{NN}} = 200$ GeV. For the evolution of the QGP produced in these collisions, we use the viscous hydrodynamic package MUSIC[18–20] with the collision-geometry-based 3D initial conditions[21] and a crossover equation of state NEOS-BQS[22] with vanishing net strangeness density $n_s = 0$ and the net electric charge-to-baryon density ratio $n_Q = 0.4n_B$. Following ref. 21, we also include in the hydrodynamic evolution a temperature and baryon chemical potential dependent specific shear viscosity $\eta/s$. At hadronization of the QGP, both hadrons and light nuclei are produced using the statistical hadronization model in the grand-canonical ensemble approach[8] with their phase-space distributions sampled on a constant energy density hypersurface according to the Cooper-Frye formula[23]. The relativistic kinetic equations for the reactions like $\pi NN \leftrightarrow \pi d$ and $\pi NNN \leftrightarrow \pi^3$H during the subsequent hadronic matter expansion are solved by a stochastic method (see "Methods" section). We further incorporate in the collision integrals the effects of finite nuclei sizes and fix the scattering cross sections for light nuclei dissociations from experimental measurements. For a hot hadronic matter in a box with periodic boundary conditions, our kinetic approach preserves the principle of detailed balance and

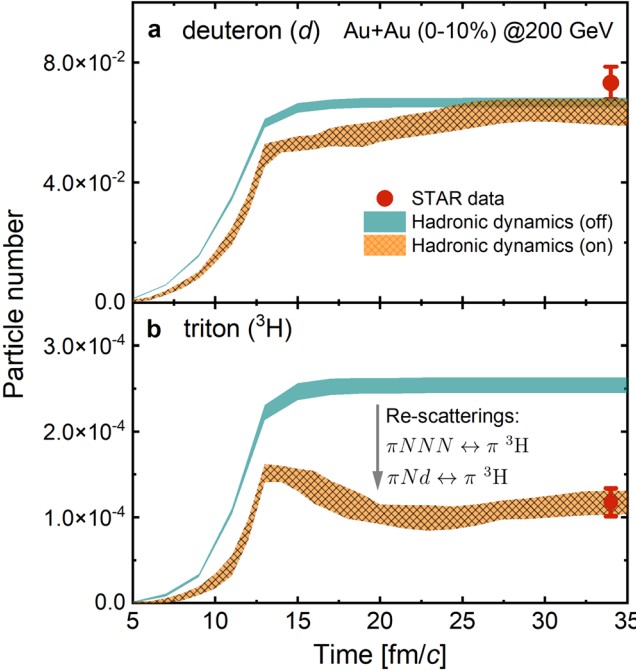

**Fig. 2 | Hadronic effects on deuteron and triton production.** Time dependence of deuteron (**a**) and triton (**b**) numbers in mid-rapidity ($|y| < 0.5$). Experimental data with combined statistical and systematic uncertainties from refs. 16,24 are denoted by filled symbols, while theoretical results with statistical uncertainties are shown by shaded bands.

reproduces the expected equilibrium light nuclei abundance. In the following, we denote by "Hydro + RKE" for this hybrid approach to the little-bang nucleosynthesis in relativistic heavy-ion collisions.

Figure 2 shows the time evolution of deuteron (upper panel (**a**)) and triton (lower panel (**b**)) numbers in mid-rapidity ($|y| < 0.5$) from theoretical calculations as well as their comparison with the experimental data from the STAR Collaboration[16,24], which are denoted by filled symbols together with their combined statistical and systematic uncertainties. The initial deuteron and triton numbers from the statistical hadronization of the QGP are shown by cyan bands, and the results after including hadronic re-scatterings using Hydro + RKE are shown by brown bands with sparse gray grids. In these collisions, the final deuteron number is about 95% of its initial value given by the SHM, which confirms the results from a recent transport model study of deuteron production[11]. The small hadronic effects on the deuteron

abundance is due to similar deuteron dissociation and regeneration rates during the hadronic evolution[11]. The hadronic effects are, however, no longer small for the final triton number. With the inclusion of $\pi NNN \leftrightarrow \pi^3 H$ and $\pi Nd \leftrightarrow \pi^3 H$ reactions, the final triton number is about half of its initial value at hadronization, which agrees well with the experimental data shown by red solid circles.

The strong hadronic re-scattering effects push the deuteron and triton chemical freeze-out to the late hadronic stage, resulting in having light nuclei only in partial chemical equilibrium during the hadronic matter expansion. This can be seen from the number density of nucleus of mass number $A$ in the high temperature and large mass limit, which is given by $\rho \approx (2J+1)(\frac{mT}{2\pi})^{\frac{3}{2}} e^{\frac{-m+A\mu}{T}}$ with $J$ and $\mu$ denoting the nucleus spin and nucleon chemical potential, respectively. For a homogeneous emission source, the equilibrium triton number ($N_{^3H}$) is related to the proton number ($N_p$) and the deuteron number ($N_d$) by $N_{^3H} \approx 0.29 N_d^2/N_p$. With the deuteron number remaining almost unchanged during the hadronic evolution, the triton number has to decrease appreciably because of the more than a factor of two increase in the proton number from resonance decays. This provides an intuitive explanation for the reduced $^3H$ number shown in Fig. 2b.

The delayed chemical freeze-out of deuteron and triton is consistent with the assumption in the nucleon coalescence model[25–27] that light nuclei are formed from a sudden recombination of kinetically freeze-out nucleons. This phenomenological model yields a similar triton number as our dynamic Hydro + RKE approach, and the close relationship between these two approaches has previously been pointed out in ref. 25. It should be mentioned, however, that deuterons are formed only from $np$ pairs in the coalescence model, while in the present kinetic approach, all $nn$, $np$, and $pp$ pairs can contribute to its production.

Similar to deuteron and triton, short-lived hadronic resonances like $K^{*0}(890)$, $\rho$, and $\Lambda(1520)$ can frequently decay and reform via e.g., $K^{*0}(890) \leftrightarrow K + \pi$, $\rho \leftrightarrow \pi\pi$, and $\Lambda(1520) \leftrightarrow N\bar{K}$. Consequently, their numbers might change as the hadronic matter expands, and such hadronic effects have indeed been observed in experiments[28,29]. Based on the exponential decay law for hadronic resonances that neglects their regenerations during the hadronic evolution, the extracted hadronic matter lifetime in relativistic heavy-ion collisions is less than 10 fm/$c$[30]. A similar short hadronic matter lifetime is found in ref. 31 using the EPOS + UrQMD model by taking the hadronic matter lifetime to be the difference between the average kinetic freeze-out time and the sum of the average hadronization time and the resonance lifetime. Using the latter definition in our Hydro + RKE model also gives a short

hadronic matter lifetime of about 9 fm/$c$. Because of the broad kinetic freeze-out time distributions in Hydro + RKE, particularly for deuterons and tritons, to properly take into account the hadronic recattering effects on their final yields requires following the time evolution of the hadronic matter to a much longer time than their mean kinetic freeze-out time as shown in Fig. 2.

## Collision energy dependence

The above calculations can be extended to other collision energies. Shown in Fig. 3 is the collision energy dependence of the light nuclei number ratios $N_d/N_p$ (a), $N_{^3H}/N_p$ (b), and $N_{^3H} \times N_p/N_d^2$ (c) in central Au + Au collisions at $\sqrt{s_{NN}} = 7.7 - 200$ GeV. For the deuteron number, both results with (Hydro + RKE) and without (SHM) hadronic effects again agree well with the experimental data within its uncertainties for all collision energies, which together with the SHM curves are both taken from ref. 16. The slightly larger $N_d/N_p$ ratio from the Hydro + RKE than the SHM prediction at $\sqrt{s_{NN}} \geq 40$ GeV is due to the smaller proton number in our calculations. For the triton number, the SHM predictions are systematically higher than the experimental data. With the inclusion of hadronic effects through our RKE approach, the triton numbers at all collision energies are reduced by about a factor of 1.8 (dashed line) and are now in good agreement with the experimental data[16]. The hadronic effects are also clearly seen in the number ratio $N_{^3H} \times N_p/N_d^2$ shown in Fig. 3c. These results provide a strong evidence for "dynamics at work"[32] in nucleosynthesis during heavy-ion collisions at all RHIC energies.

In the STAR data, the energy dependence of $N_{^3H} \times N_p/N_d^2$ shows a possible enhancement at $\sqrt{s_{NN}} \approx 20-30$ GeV, and this behavior could be due to the appearance of a critical point in the QCD phase diagram[33–36]. Since the transition from the QGP to the hadronic matter in our model calculations is always a smooth crossover, no critical dynamics is expected. The resulting energy dependence of $N_{^3H} \times N_p/N_d^2$ in our study is thus essentially flat as in previous studies using the coalescence model based on kinetically freeze-out nucleons from hybrid or transport models[27,37].

It is worthwhile to mention that for Au + Au collisions at lower energies ($\sqrt{s_{NN}} \leq 20$ GeV), the hadronic matter becomes more baryon rich, and hadronic re-scatterings with the nucleon as a catalyzer for light nuclei production is expected to play an increasingly important role[38]. We find, however, that the resulting enhanced scattering rates for deuteron and triton production only have small effects on their final numbers even at the low collision energy of 7.7 GeV when the nucleon density becomes non-negligible compared to the pion density. This is mainly because both dissociation and regeneration rates

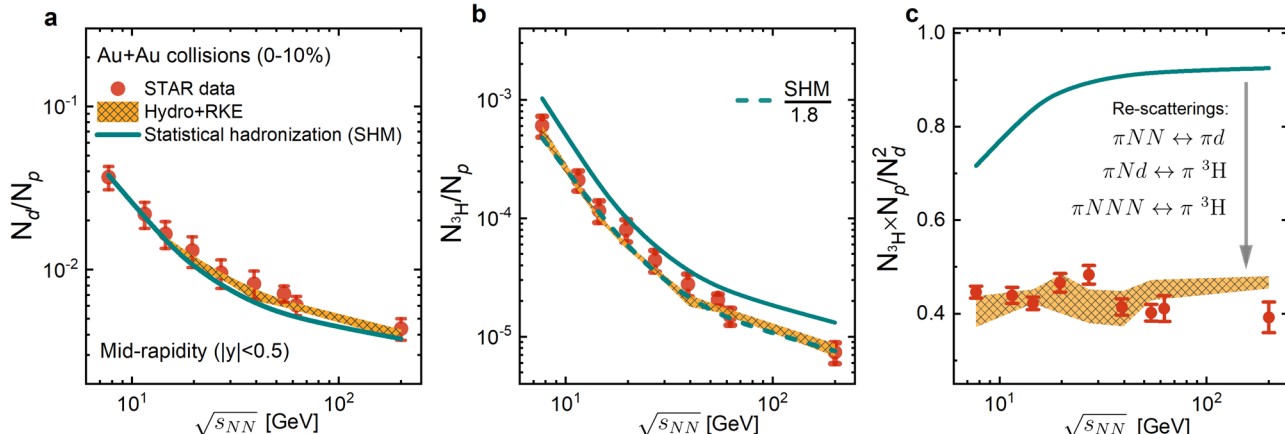

**Fig. 3 | Deuteron and triton production in Au+Au collisions at RHIC energies.** Collision energy dependence of hadronic re-scattering effects on light nuclei yield ratios $N_d/N_p$ (**a**), $N_{^3H}/N_p$ (**b**), and $N_{^3H} \times N_p/N_d^2$ (**c**). Theoretical results with and without hadronic dynamics are from Hydro + RKE (shaded bands) and SHM[8,16] (lines), respectively. Experimental data points with combined statistical and systematic uncertainties are from the STAR Collaboration[16].

are enhanced at a similar level, resulting in an almost complete cancellation of their effects.

### Hadronic effects in Pb + Pb collisions at the LHC energies

Similar hadronic effects are also present in Pb + Pb collisions at the LHC energies. Figure 4 displays the charged-particle multiplicity dependence of light nuclei number ratios $N_d/N_p$ (a), $N_{^3\text{He}}/N_p$ and $N_{^3\text{H}}/N_p$ (b). Experimental data denoted by symbols are from the ALICE Collaboration[17,39]. For the $N_d/N_p$ ratio, the inclusion of hadronic re-scatterings has only a very small effect as in Au + Au collisions at RHIC energies. For $^3$He and $^3$H production, the SHM prediction again significantly overestimates the new data (solid stars and circles) at $\sqrt{s_{NN}}$ = 5.02 TeV[17], although it agrees with the single data point (gray square) for $N_{^3\text{He}}/N_p$ measured at $\sqrt{s_{NN}}$ = 2.76 TeV[39]. With the inclusion of pion-catalyzed multi-body reactions, our model calculation reproduces well the newly measured $N_{^3\text{He}}/N_p$ and $N_{^3\text{H}}/N_p$ at $\sqrt{s_{NN}}$ = 5.02 TeV. The much higher statistics in the new measurements at $\sqrt{s_{NN}}$ = 5.02 TeV allow us to conclude that the strong hadronic re-scattering effects found in our calculation are in good agreement with measurements at beam energies across three orders of magnitude ($\sqrt{s_{NN}}$ = 7.7 − 5020 GeV).

We notice that the reduced $^3$H number due to the hadronic effects in our results is absent in a recent study using the rate equation[10] in a simple isentropic expansion model for the hadronic matter evolution. In this schematic approach, both kinetic thermal equilibrium and isentropic expansion are assumed for the hadronic matter when solving the rate equations for light nuclei dissociation and regeneration. These assumptions become questionable near the kinetic freeze-out when the system is driven out of chemical equilibrium. As a result, the

absence of hadronic re-scattering effects obtained in this approach leads to results that are in disagreement with the latest measurements on the triton and helium-3 numbers at RHIC and the LHC.

In the above calculations, we have only considered central collisions where canonical effects due to charge conservation at hadronization of the QGP are small[40]. Extending our study to peripheral collisions by including the canonical effects at hadronization is of great interest for future studies.

## Discussion

The present study shows via the relativistic kinetic approach that the post-hadronization dynamics plays an important role in the little-bang nucleosynthesis during high-energy heavy-ion collisions. We find, in particular, that the triton number produced from the created QGP in central Au + Au collisions is reduced by about a factor of 1.8 during the subsequent hadronic matter expansion, although the deuteron number is essentially not affected. These distinct hadronic effects on deuteron and triton production are in excellent agreement with recent measurements by the STAR Collaboration at RHIC and are further supported by the latest measurement by the ALICE Collaboration at the LHC. Our study thus shows the inadequacy of the statistical hadronization model for understanding triton production in these collisions. In contrast to the big-bang nucleosynthesis, in which photonuclear reactions dominate its dynamics, our model-to-data comparison unveils the importance of pion-catalyzed multi-body reactions on the dynamics of the little-bang nucleosynthesis in relativistic heavy-ion collisions.

## Methods

### Relativistic kinetic equations

We illustrate our approach by considering the specific channel $\pi^+ d \to \pi^+ np$ for deuteron dissociation. With the typical temperature of the hadronic matter in high-energy nuclear collisions at 100–150 MeV, the pion thermal wavelength is around 0.4–0.5 fm, which is much smaller than the deuteron diameter or size of about 4 fm. A pion thus has a sufficiently large momentum to resolve the two constituent nucleons in a deuteron, resulting its scattering by one nucleon with the other nucleon being a spectator, as shown in Fig. 5. This quasifree or impulse approximation (IA) was previously used in studying deuteron dissociation in low-energy heavy-ion collisions[12,41] and also $J/\Psi$ dissociation by partons in relativistic heavy-ion collisions[42]. Under this approximation, the inverse reaction $\pi^+ np \to \pi^+ d$ of pion-catalyzed deuteron production from two nucleons can be viewed as a two-step process, i.e., the scattering between the pion and a nucleon with the final-state nucleon sightly off the mass shell and then subsequently coalescing with the spectator nucleon to form the deuteron.

Specifically, the production and dissociation of deuterons in a pion-dominated hadronic matter can be similarly described by the relativistic kinetic equation (RKE) that was previously derived in ref. 12 for deuteron production from the reactions $NNN \leftrightarrow Nd$ in a pure

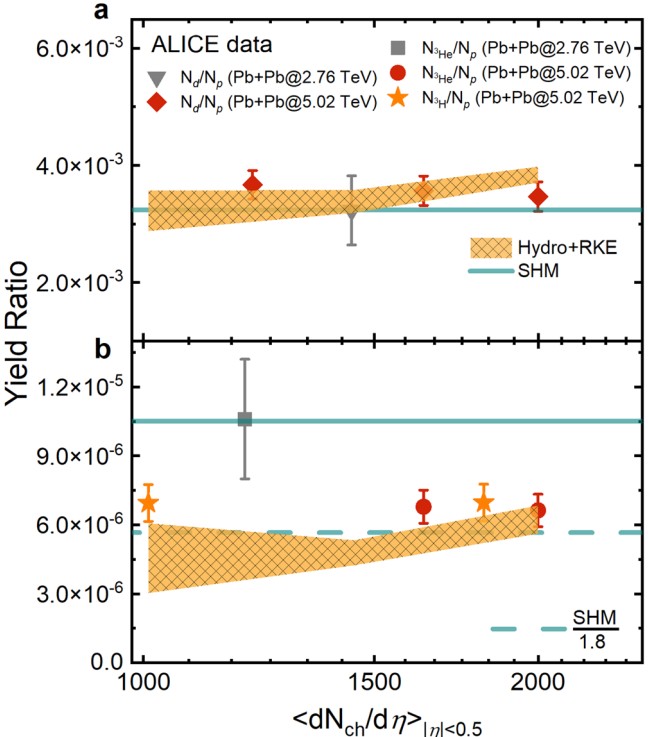

**Fig. 4 | Deuteron, triton, and helium-3 production in Pb+Pb collisions at the LHC energies.** Charged-particle multiplicity ($dN_{ch}/d\eta$) dependence of hadronic re-scattering effects on light nuclei yield ratios $N_d/N_p$ (**a**), $N_{^3\text{He}}/N_p$ and $N_{^3\text{H}}/N_p$ (**b**). Theoretical results with and without hadronic dynamics are from Hydro + RKE (shaded bands) and SHM[8,16] (lines), respectively. Experimental data points with combined statistical and systematic uncertainties are from the ALICE Collaboration[17,39].

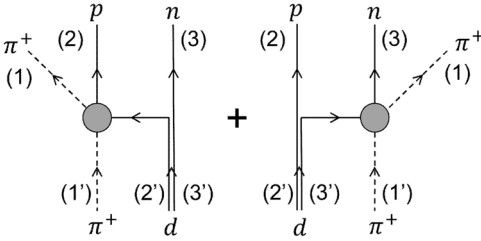

**Fig. 5 | Diagrams for the reaction $\pi^+ d \leftrightarrow \pi^+ np$ in the impulse approximation.** The filled bubble indicates intermediate states such as the $\Delta$ resonance.

nucleonic matter, i.e.,

$$\frac{\partial f_d}{\partial t} + \frac{\mathbf{P}}{E_d} \cdot \frac{\partial f_d}{\partial \mathbf{R}} = -\mathcal{K}^> f_d + \mathcal{K}^< (1 + f_d), \qquad (1)$$

where the deuteron distribution function $f_d(\mathbf{R}, \mathbf{P})$ is normalized as $g_d(2\pi)^{-3}\int d^3\mathbf{R} d^3\mathbf{P} f_d = N_d$ with $g_d = 3$ and $N_d$ denoting the deuteron spin degeneracy factor and number, respectively. On the l.h.s. of Eq. (1), which corresponds to the drift term, we treat the deuteron as a free particle by neglecting its mass change and mean-field potential. The two terms on the r.h.s. of Eq. (1) describe the deuteron dissociation ($\mathcal{K}^>$) and regeneration ($\mathcal{K}^<$) rates, respectively. Under the impulse approximation, the collision integral on the r.h.s. is given by[12,43]

$$\frac{1}{2g_d E_d} \int \prod_{i=1}^{3} \frac{d^3\mathbf{p}_i}{(2\pi)^3 2E_i} \frac{d^3\mathbf{p}_\pi}{(2\pi)^3 2E_\pi} \frac{E_d d^3\mathbf{r}}{m_d}$$
$$\times 2m_d W_d(\tilde{\mathbf{r}}, \tilde{\mathbf{p}})(|\mathcal{M}_{\pi^+ n \to \pi^+ n}|^2 + n \leftrightarrow p)$$
$$\times \left[ -\left( \prod_{i=1}^{3}(1 \pm f_i) \right) g_\pi f_\pi g_d f_d + \frac{3}{4}\left( \prod_{i=1}^{3} g_i f_i \right) \right.$$
$$\left. \times (1 + f_\pi)(1 + f_d) \right] \times (2\pi)^4 \delta^4(p_{\text{in}} - p_{\text{out}}), \qquad (2)$$

from which $\mathcal{K}^>$ and $\mathcal{K}^<$ in Eq. (1) can be identified. In the above equation, the factor $1 \pm f_i$ in the square bracket is due to the quantum statistics of fermions (−) and bosons (+), and the $\delta$-function denotes the conservation of energy and momentum with $p_{\text{in}} = \sum_{i=1}^{3} p_i$ and $p_{\text{out}} = p_\pi + p_d$. The factor 3/4 in the third line is due to the spin factors of initial and final states. The second line of Eq. (2) is the spin-averaged squared amplitude between two scattering nucleons that are separated by a distance $\mathbf{r}$, resulting in a nonlocal collision integral. The $W_d$ denotes the deuteron Wigner function, with $\tilde{\mathbf{r}} = \tilde{\mathbf{r}}_n - \tilde{\mathbf{r}}_p$ and $\tilde{\mathbf{p}} = (\tilde{\mathbf{p}}_n - \tilde{\mathbf{p}}_p)/2$ being, respectively, the relative coordinate and momentum in the center-of-mass frame of the neutron and proton that forms the deuteron. For simplicity, we take $W_d = 8e^{-\tilde{\mathbf{r}}^2/\sigma_d^2 - \tilde{\mathbf{p}}^2 \sigma_d^2}$ with $\sigma_d = 3.2$ fm to reproduce the empirical deuteron root-mean-squared radius of $r_d = 1.96$ fm[25].

For an approximately uniform system, the spatial part of $W_d$ in Eq. (2) can be integrated out, leading to $|\phi_d(\tilde{\mathbf{p}})|^2 = \int d^3\mathbf{r} \gamma_d W_d = (4\pi\sigma_d^2)^{3/2} e^{-\tilde{\mathbf{p}}^2 \sigma_d^2}$ with $\gamma_d = E_d/m_d$. This reduces the second line of Eq. (2) to $2m_d|\phi_d|^2(|\mathcal{M}_{\pi^+ p \to \pi^+ p}|^2 + p \leftrightarrow n)$, which is the usual impulse approximation for $|\mathcal{M}_{\pi^+ d \to \pi^+ np}|^2$, e.g., adopted in ref. 12.

The pion-nucleon scattering matrix element $\overline{|\mathcal{M}_{\pi^+ p \to \pi^+ p}|^2}$ can be related to the $\pi N$ scattering cross section[44]. Under the impulse approximation, the deuteron dissociation cross section by a

pion of large momentum ($p_{\text{lab}}$) is approximately given by $\sigma_{\pi^+ d \to \pi^+ np} \approx \sigma_{\pi^+ n \to \pi^+ n} + \sigma_{\pi^+ p \to \pi^+ p} \approx \frac{4}{3}\sigma_{\pi^+ p \to \pi^+ p}$[12,45]. For a low momentum pion, e.g., 0.3 GeV, a renormalization factor $F_d$[12] is, however, needed, i.e., $\sigma_{\pi^+ d \to \pi^+ np} = F_d(\sigma_{\pi^+ n \to \pi^+ n} + \sigma_{\pi^+ p \to \pi^+ p})$. As shown in Fig. 6a, b, using the constant values $F_d \approx 0.72$ and $F_{^3\text{He}} \approx 0.51$ leads to an excellent description of the data[45–51] for the $\pi + d$ and $\pi + {}^3\text{He}$ dissociation cross sections in the energy regime relevant for the present study.

## Stochastic method

To numerically solve Eq. (1) with the nonlocal collision integral given by Eq. (2), we adopt the test particle ansatz[52], i.e., mimicking the distribution functions $f_\alpha$ of a certain particle species of number $N_\alpha$ by a large number of delta functions, $f_\alpha(\mathbf{r}, \mathbf{p}) \approx (2\pi)^3/(g_\alpha N_{\text{test}}) \sum_{i=1}^{N_\alpha N_{\text{test}}} \delta(\mathbf{r}_i - \mathbf{r})\delta(\mathbf{p}_i - \mathbf{p})$. The $g_\alpha$ and $N_{\text{test}}$ denote the spin degeneracy factor and the number of test particles, respectively. The following stochastic method[13,53,54] is then used to evaluate the collision integrals. To ensure the convergence of numerical results, a sufficiently large $N_{\text{test}}$ needs to be used.

Given the deuteron dissociation and regeneration rates by a pion from Eq. (2), the probability for the reaction $\pi^+ d \to \pi^+ np$ between a pion and a deuteron inside a volume $\Delta V$ to take place within a time interval $\Delta t$ is then given by[12,53,54]

$$P_{23}\Big|_{\text{IA}} \approx F_d v_{\pi^+ p} \sigma_{\pi^+ p \to \pi^+ p} \frac{\Delta t}{N_{\text{test}} \Delta V} + (p \leftrightarrow n), \qquad (3)$$

where $v_{\pi^+ p}$ is the relative velocity between the pion and the proton inside the deuteron, and the two terms on the r.h.s correspond, respectively, to the two diagrams in Fig. 5. Similarly, the probability for the reaction $\pi^+ np \to \pi^+ d$ to take place inside a volume $\Delta V$ in a time interval $\Delta t$ is

$$P_{32}\Big|_{\text{IA}} \approx \frac{3}{4} F_d v_{\pi^+ p} \sigma_{\pi^+ p \to \pi^+ p} \frac{\Delta t W_d}{N_{\text{test}}^2 \Delta V} + (p \leftrightarrow n). \qquad (4)$$

Note that the deuteron Wigner function $W_d$ in Eq. (4) encodes the nonlocality of the scattering as it depends on both the coordinates and momenta of the constituent nucleons inside the deuteron. It can be replaced by $|\phi_d|^2/(\gamma_d \Delta V)$ if the volume of the hadronic matter is much larger than the deuteron size.

The above treatment for deuteron production and dissociation can be straightforwardly generalized to ${}^3\text{H}$ production from the reactions $\pi NNN \leftrightarrow \pi {}^3\text{H}$ and $\pi Nd \leftrightarrow \pi {}^3\text{H}$, and similarly for ${}^3\text{He}$. For the $3 \leftrightarrow 2$ reaction, its treatment is similar to that for deuteron production. The probability for the $4 \to 2$ reaction to occur in a volume $\Delta V$ within a time

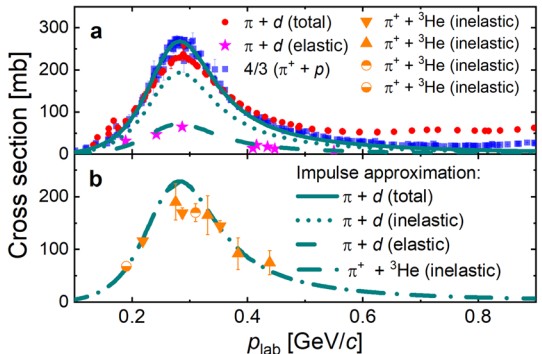

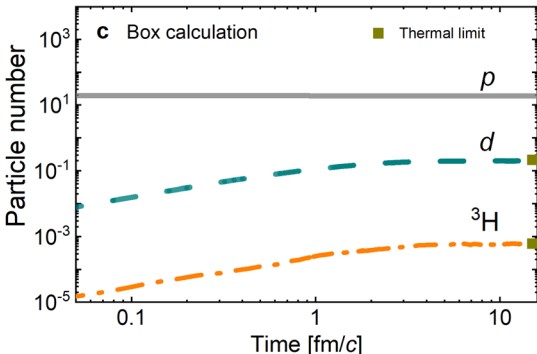

**Fig. 6 | Cross sections and thermal limit.** Comparison of the pion incident momentum ($p_{\text{lab}}$) dependence of the cross sections for the reactions $\pi d$ (**a**) and $\pi {}^3\text{He}$ (**b**) between experimental data[45–51] with combined statistical and systematic uncertainties and our fit based on the impulse approximation (lines). Box calculations for light nuclei production from our kinetic approach (**c**).

interval $\Delta t$ is, however, given by

$$P_{42}\Big|_{\mathrm{IA}} \approx \frac{1}{4} F_{^3\mathrm{H}} \frac{v_{\pi N} \sigma_{\pi N \to \pi N} \Delta t}{N_{\mathrm{test}}^3 \Delta V} W_{^3\mathrm{H}}, \tag{5}$$

where $F_{^3\mathrm{H}} \approx F_{^3\mathrm{He}} \approx 0.51$, and the triton Wigner function $W_{^3\mathrm{H}}$ is also taken to have Gaussian forms as in refs. 25,55. The branching ratio for the dissociation of $^3$H into $\pi Nd$ is about 0.56, which is obtained from the overlap between the deuteron wave function and the wave function of the two nucleons inside the triton.

## Thermal limit in box calculation

To validate the above stochastic method, we consider deuteron and triton production in a $(20 \text{ fm})^3$ box with periodic boundary conditions. Initially, 24 protons, 24 neutrons, and 480 pions for each of its three charge states are uniformly distributed in the box, corresponding to an almost vanishing chemical potential $\mu_B \approx 0.007$ GeV at top RHIC and the LHC energies. Their initial momentum distributions are taken to have the thermal Boltzmann form with temperature $T = 155$ MeV. The right panel of Fig. 6 shows the time evolution of the $p$, $d$, and $^3$H numbers. It is seen that their final numbers are consistent with their expected thermal values at chemical equilibrium over five orders of magnitude, suggesting that the detailed balance is well preserved in our calculations.

## Data availability

All the data supporting the findings in this work are available within the manuscript and any additional data are available from the corresponding author upon request.

## Code availability

Inquiries about the code in this work will be responded to by the corresponding authors.

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

## Acknowledgements

We thank Lie-Wen Chen and Xiaofeng Luo for fruitful discussions, and Chen Zhong for the maintenance of the CPU and GPU clusters. This work was supported in part by the National Natural Science Foundation of China under contract No. 12375121, No. 11891070, No. 11890714, No. 12147101, the National Key Research and Development Project of China under Grant No. 2022YFA1602303, the 111 Project, Guangdong Major Project of Basic and Applied Basic Research No. 2020B0301030008, the U.S. Department of Energy under Award No. DE-SC0015266, No. DE-SC0013460, No. DE-SC0021969, and the National Science Foundation under grant number PHY-2012922.

## Author contributions

Kai-Jia Sun and Rui Wang carried out all numerical calculations and contributed equally to this work. Che Ming Ko, Yu-Gang Ma, and Chun Shen engaged in the development of the theoretical framework and the preparation of the manuscript.

## Competing interests

The authors declare no competing interests.
