## [Peer Review File · Nature Communications]

Unveiling the dynamics of little-bang nucleosynthesisREVIEWER COMMENTS

Reviewer #1 (Remarks to the Author):

The authors study the production of light nuclei, i.e. deuterons and tritons in relativistic nuclear collisions. To this aim they compare two different calculations, one based on relativistic kinetic theory and one based on a statistical hadronization approach with available experimental data up to the top RHIC energy (200 GeV).

The main finding is that the production of tritons can not be well described using a statistical hadronization approach (which overpredicts the data by a factor of 1.8), but a kinetic theory calculation (i.e. a fully dynamical transport simulation) is needed to explain the triton data. This finding is certainly important, because it indicates the application limits of the statistical hadronization approach to particles emitted directly from the chemical freeze-out hypersurface and shows that such an approach may not be used for light clusters or e.g. hadron resonances that are emitted late.

The paper is generally well written, the methodology is clear and the analysis is sound. However some questions remain that should be addressed by the authors:

1) It is not really clear how the statistical model curves in Fig. 2 for deuterons and tritons at different times have been obtained. Are they obtained fits to the full amount of hadron yields at each time? Are they taken from the Cooper-Fry (CF) prescription at a hypersurface of fixed energy density? What kind of EoS has been used to translate energy/baryon density to temperature/chem. potential (there are various NEOS-EoS)?

2) Is the baryon number in the CF-transition conserved locally or globally? This might introduce additional correlations among the baryons, e.g. local conservation might lead to additional annihilation of baryons in the kinetic equation.

3) How have the statistical model results been restricted to midrapidity in Fig.3? I am asking because different methods can be used. E.g. one could integrate over all momenta and take the ratios afterwards, however, using the (T, μ) extracted at midrapidity (often used but not correct) or one can use the differential thermal distribution incl. the longitudinal momentum dependence and restrict the integration to the region around midrapidity (rarely used, but more correct). Both approaches yield different results for the ratios due to the longitudinal flow. They should also clarify which flow profile was used.

4) The authors have omitted the data point at LHC. This is important because the cluster yields at LHC can be described by the statistical model and the RHIC data seems to be consistently lower than the LHC data. To make the point (as the authors do) that the RKE is needed to describe the data their approach should also be validated against the LHC data to allow for a firm conclusion.

5) The results strongly depend on the total pion yield and how the pions are transported during the time evolution of the system. Because it might be that the pions are traveling in rho-mesons and do not participate in the reactions described in the paper or do the authors oversaturate they pion density because the pions are considered already free? The authors should therefore demonstrate that their final pion yield is compatible with the experimental results and should clarify what kind of pions are used.

5) The connection to similar observations made from hadronic resonances (K^* ,...) should be discussed.

To summarize, the idea and the findings are interesting, but at the moment further information is needed before a conclusion can drawn.

Reviewer #2 (Remarks to the Author):

The manuscript presents a study of the production of light nuclei (deuteron, triton) in collisions of nuclei at large energies. It is a very relevant subject and the approach presented in the manuscript very good in principle and interesting, but I find that its strong claims are not backed by equally strong proofs.

The authors claim to explain the relative production of deuteron and triton nuclei to that of protons over a broad collision energy (per nucleon pair, $\sqrt{s_{NN}}$) range from about 8 to 200 GeV. The author's approach is to start from the predictions of the statistical hadronization model (SHM), which describes hadron production at the instant of the (sudden) chemical freeze-out (at a temperature of about 156 MeV, for the collision range explored here) and follow the dynamics of the system in its hadronic stage. The conclusion is that the predictions by SHM for the deuterons are not affected by the hadronic stage, while a strong reduction is observed for the tritons, in agreement with the data measured by the STAR collaboration.

Several similar studies have been published already, refs. [9,10,11], focusing on the hadronic stage at LHC energies (2.76 TeV). In ref. [10], no effect is seen in the hadronic stage for both deuterons and tritons (while that study is in agreement with the ALICE data). The authors note this difference in the present manuscript, but give no clarification at all. The authors of the present manuscript have shown in an

earlier manuscript posted on arXiv (2106.12742, sharing some Figures with the present manuscript) a similarly good description of the data at the LHC (including also pp collisions).

And here lies my problem: I cannot see why there should be no effect at the LHC energies and a significant effect at lower energies, given that the hadronic system is at the energy of 200 GeV very similar to that of 2.76 TeV (with the exception of a small asymmetry in favor of matter compared to antimatter, but this cannot play a major role).

In the similar approach for the deuterons published in ref. [11] (which the authors cite properly), focused at the larger energy of the LHC where the pions dominate, it is pointed out the importance of the nucleons for the $d+X \rightarrow np$ reactions. Since at the lower energies explored in the present manuscript the proton/pion ratio is around 0.5 (at $\sqrt{s_{NN}}=8$), the channels involving nucleons ($X=n,p$) cannot be neglected. They are characterized by cross sections of similar magnitude as the $\pi+d$ reactions (see ref.[11]) and are consequently expected to bring a significant contribution. This is another major shortcoming of the present manuscript.

Sure, the question remains why SHM explains the ^3He yield measured by ALICE at 2.76 TeV and not the measurements on tritons by STAR (ref. 16]), but the observation in the present manuscript of a large effect in the hadronic stage needs stronger proofs. In particular, the detailed balance for the reaction channels relevant for the tritons needs to be demonstrated (the authors of refs. 9, 10, 11 put, rightly so, a great emphasis on this).

Further aspects which I find weak in the present manuscript:

The time scales needed for the equilibration in the hadronic stage shown in Fig. 2 are very long, implying a hadronic stage of about 25 fm/c. To me, this is very unrealistic, in particular since the measurements at the LHC (ALICE, arXiv:2211.04384) limit the hadronic phase to 10 fm/c and at lower energies the duration has to be smaller.

In Fig. 3, left plot, one observes an increase of the d/p ratio compared to the SHM prediction, it is not clear how this can happen, given the results shown in Fig. 2.

Thermal limit in a box: 480 pions of each charge and 24 protons and neutrons are used, corresponding to the energy of 200 GeV (roughly at least), for one unit of rapidity. But the volume of 1 unit of rapidity at chemical freeze-out (which describes the abundancies quoted above) is about 2000 fm^3 , while the authors use here (strangely and without any justification) 8000 fm^3 !

No explanation is given on the uncertainty bands shown in Fig. 2 and 3.

Reviewer #3 (Remarks to the Author):

The article titled "Unveiling the dynamics of little-bang nucleosynthesis" investigates the production of light nuclei from the quark-gluon plasma (QGP) through the statistical hadronization model. Specifically, the article investigates the mechanism of hadronic re-scattering and its effect on equilibrium light nuclei yields. The research direction is timely, motivated by the recent measurement by the STAR collaboration reporting the triton production over a wide range in collision energies. The precise production mechanism of light nuclei in heavy-ion collisions has been a topic of great debate and has therefore garnered significant experimental and theoretical interest in recent years. The contribution here potentially makes an important stride towards understanding the light nuclei production mechanism.

This article addresses topics of interest in high energy nuclear physics with implications for the interpretation of light nuclei production in heavy-ion collision experiments.

The studies presented in this article appear to be high quality with appropriate data and methods provided to clearly communicate the claims in the paper. However, based on the questions/comments listed below I find that the level of support for the primary conclusions could be improved through additional checks and comparisons with existing experimental data. The results are significant and make important progress in understanding the production mechanisms leading to the final yield of light nuclei in high-energy heavy-ion collisions a topic of intense interest. The methods employed and the presentation of their results are clear with a few exceptions – see the questions/comments below related to the data presentation and the uncertainty on the theoretical curves. Overall, I find the article well written and organized. My main concern with the article and the conclusions relates to the ability to describe the LHC data at higher energies (2.76 TeV) and the multiplicity dependence of the $N_{3H} \cdot N_p / N_d^2$ data. The article may be suitable for publication if these points can be adequately addressed.

1. A few key statements come across too strong

pg2 rhs middle paragraph: "This effect has indeed been observed in recent experimental measurements at the Relativistic Heavy Ion Collider (RHIC)"

and in the conclusions:

"These distinct hadronic effects on deuteron and triton production have been confirmed in recent measurements by the STAR collaboration at RHIC."

I find these statements too strong, since coalesce and transport models are also in good agreement. What this article does not seem to address is any way of distinguishing these models as the true mechanism (or otherwise unifying their interpretation in some way, see my note below). Some comment on this consistency with respect to the coalesce model is given on page 4 lhs middle.

More neutral statement like the one just before "Results and Discussion" seem more appropriate - i.e. that the existing STAR results are in excellent agreement with this prediction:

"However, the initial triton yield from hadronization of the QGP is reduced by about a factor of 1.8 during the hadronic matter expansion, leading to a result that is in excellent agreement with the latest measurements by the STAR Collaboration [16]."

In my opinion these statements would leave a reader without additional domain specific knowledge with the impression that the exact mechanism of light nuclei is established with no debate. This seems to be over stating the situation since, for instance the experimental data for light nuclei yields can be equally well explained by coalescence or hadronic transport models. If the authors mean by page 4 lhs middle that this finding unites or unifies the interpretation of these models then it should be made a more specific claim/statement of the article.

2. STAR data:

- What exactly is plotted on Fig. 2 with respect to the STAR datapoint from [16] - as I could not find points in the STAR manuscript corresponding exactly to these values? Are they integrated yields (over pT), and if so is there some extrapolation uncertainty? Do the datapoint's error bars show combined statistical and systematic uncertainties, if so that should be noted somewhere in the caption of Fig. 2 (and for other figures).

3. Multiplicity dependence

Fig. 3 of the STAR triton paper [16] shows that the thermal model has a totally incorrect dependence on multiplicity ($dN_{ch} / d\eta$) compared to data. The article discusses the impact of the hadronic re-scattering with comparisons in the 0-10% most central collisions where the authors note the excellent agreement. But what about other centralities (lower particle multiplicities) and generally if the effect leads to an accurate multiplicity dependence? If not is there a justification for this?

4. Source of uncertainty bands and some discussion

In Fig. 2 and 3 the results are shown as shaded bands but there is no discussion of the source of spread (uncertainty) in the calculation. Since the SHM curve appears to be a line (not a band) it is presumably from the statistical error on the stochastic integration? It should be commented as to what contributes and if any other model uncertainties are included in this band.

This uncertainty becomes especially interesting and important with respect to the discussion of the QGP critical point search where the uncertainty on the 'almost flat' prediction is of the same order as the uncertainty on the data itself.

5. Consistency with the LHC measurements

from the STAR triton paper [16] :

"Quantitatively, the thermal model describes the Nd/Np ratios well, but it systematically overestimates the Nt/Np ratios except for the results from central Pb + Pb collisions at $\sqrt{s_{NN}} = 2.76$ TeV"

Based on this discussion and Fig. 3 in this article it seems that these hadronic re-scattering effects have a nearly beam-energy independent effect on $N(3H) / N_p$ - reducing the SHM yield ~uniformly by a factor of 1.8.

In fact the article says:

"As can be inferred from Fig. 3, similar hadronic effects on triton production are expected at both lower and higher beam energies than those covered in the RHIC Beam Energy Scan program."

Some comment should be made about the LHC measurement at 2.76 TeV. If the existing thermal model results are already in good agreement with the measurement at that energy it would imply that this dynamical effect would lead to an underestimation of the yield measured at 2.76 TeV. The article discusses the energy dependence, so presumably the comparison can be extended to these higher energy collisions? I would be interested in seeing such a comparison. Regardless of whether or not it is added to the Fig 3. it could be commented on in the article.

6. Clarity about the key difference of this approach.

pg.2 rhs: "the post-hadronization dynamics has only small effects on their final abundance. This result is supported by studies using the Saha equation [9] and the rate equation [10] as well as the more microscopic transport approach [11]."

page 4 lhs: "However, the decreased 3H yield due to the hadronic effects in our results is almost absent in a recent study using the rate equation [10] in a simple isentropic expansion model for the hadronic matter evolution."

similarly in the STAR paper (references refer therein):

"Dynamical model calculations with hadronic rescatterings implemented using both the saha [42] and rate equations [66] show that the deuteron, triton, and helium-3 yields remain unchanged during hadronic expansion."

and the article comments on this in p2 rhs top paragraph. However, the reader is left wondering how two completely different results can be found for the effect of hadronic re-scattering (those quoted above) and that found in this article? It would be helpful to comment on the potential source of this discrepancy.

Finally I have a few specific comments on wording and/or grammar that could improve the flow and readability of the manuscript:

“and etc.” - maybe provide one more example and drop the “and etc.” either way it should be “etc.” (without and)

“is a smooth crossover located at $T = 156.5$ ” – located only makes sense if you also say “in the phase diagram”, consider “... characterized by $T = \dots$ ”

“Analyses of experimental data based on this statistical hadronization model” – “this”->“the”

“rescattering” vs. “re-scattering” be consistent

In several places “the” is used when it should be omitted and vice versa, e.g.:

“The small hadronic effects on the deuteron abundance is due to similar deuteron dissociation and regeneration rates during the hadronic evolution” -> “... hadronic evolution” (remove the)

“The equilibrated triton number (N_{3H}) can then be related to the proton number (N_p) and deuteron number (N_d)” -> missing “the” -> “.. and the deuteron number ... “

“the inclusion of hadronic effects, the triton yields at all collision energies” -> “..., triton yields at all collision energies”

“its greater sensitivity to the hadronic dynamics than nucleons and deuterons” -> “its greater sensitivity to hadronic dynamics than nucleons and deuterons” (remove the)

“As to the triton yield” -> “For the triton yield”

Reply (NCOMMS-23-24526A)

We thank the reviewers for their insightful and constructive reports on our manuscript. Following their suggestions, we have made extensive revisions on the structure, format, presentation, and analysis of our findings, which we hope has led to a manuscript the reviewers will find it greatly improved and having a more accurate and reliable conclusion.

Among the many changes we have made on the manuscript, we would like to specifically mention our response to the concern from the reviewers about the reliability and generalizability of our conclusion on the importance of hadronic dynamics in the little-bang nucleosynthesis. To this end, we have extended our study to higher beam energies and validated our results against the latest experimental measurements from the LHC. Also, we have addressed the reviewers' concern about several technical issues by adding additional detailed explanations. Moreover, we have updated the overwhelming evidence from both experimental measurements and theoretical studies for the vital role played by the hadronic dynamics in the little-bang nucleosynthesis during high-energy nuclear collisions. We believe that the general public and physics community will be fascinated by our findings on the dynamics of little-bang nucleosynthesis in our improved manuscript.

Please find below our response to each question of the reviewers and the related changes we have made in our manuscript.

REVIEWER 1

Introductory comment:

The authors study the production of light nuclei, i.e. deuterons and tritons in relativistic nuclear collisions. To this aim they compare two different calculations, one based on relativistic kinetic theory and one based on a statistical hadronization approach with available experimental data up to the top RHIC energy (200 GeV).

The main finding is that the production of tritons can not be well described using a statistical hadronization approach (which overpredicts the data by a factor of 1.8), but a kinetic theory calculation (i.e. a fully dynamical transport simulation) is needed to explain the triton data. This finding is certainly important, because it indicates the application limits of the statistical hadronization approach to particles emitted directly from the chemical freeze-out hypersurface and shows that such an approach may not be used for light clusters or e.g. hadron resonances that are emitted late.

The paper is generally well written, the methodology is clear and the analysis is sound.

Authors:

We appreciate the reviewer's insightful and constructive comment and advice, and we have carefully addressed these concerns and made a proper revision of the manuscript. These

comments and suggestions have not only enabled us to provide a highly improved manuscript but also inspired us to conduct more in-depth studies on the production of light (hyper)nuclei and hadronic resonances in future works. In particular, we have extended our study from RHIC energies ($\sqrt{s_{NN}} = 7.7 - 200$ GeV) to higher LHC energies ($\sqrt{s_{NN}} = 5.02$ TeV) and pointed out that the triton and helium-3 yields recently measured in Pb+Pb collisions ($\sqrt{s_{NN}} = 5.02$ TeV) by the ALICE Collaboration are also overestimated by the statistical hadronization model (SHM). Similar to the situation in Au+Au collisions at RHIC energies, we have also shown that this overestimation can be understood and resolved after including multi-body pion-catalyzed reactions in our kinetic approach. Consequently, our conclusion is further strengthened by the experimental measurements at the LHC.

Comment #1:

However some questions remain that should be addressed by the authors:

1) It is not really clear how the statistical model curves in Fig. 2 for deuterons and tritons at different times have been obtained. Are they obtained fits to the full amount of hadron yields at each time? Are they taken from the Cooper-Fry (CF) prescription at a hypersurface of fixed energy density? What kind of EoS has been used to translate energy/baryon density to temperature/chem. potential (there are various NEOS-EoS)?

Authors: In the present Hydro+RKE approach, particles including light nuclei are initially produced at the hadronization or particlization stage. Their numbers are taken from the Cooper-Fry (CF) prescription at a hypersurface of fixed energy density. Specifically, we convert fluid cells to hadrons using the Monte-Carlo sampling based on the Cooper-Fry formula when their energy densities drop below the switching energy density $e_{sw}=0.26$ GeV/fm³. Every produced particle is then assigned an identification vector array in the form of (id, px, py, pz, mass, x, y, z, time). In each time step, we count the deuteron and triton numbers in mid-rapidity and monitor how their values evolve as a function of time.

In addition, we have used a crossover type of equation of state (EoS) (NEOS-BQS) with the strangeness neutrality condition of vanishing net strangeness density, $n_s = 0$, and the net electric charge-to-baryon density ratio $n_Q = 0.4n_B$ [Phys. Rev. C100, 024907 (2019)]. Following Ref. [Phys. Rev. C102, 014909 (2020)], we include in the hydrodynamic evolution a temperature and baryon chemical potential dependent specific shear viscosity η/s .

Changes made in response to comment #1: On page 3, left column, we have added the sentence, “with vanishing net strangeness density $n_s = 0$ and the net electric charge-to-baryon density ratio $n_Q = 0.4n_B$. Following [Phys. Rev. C 102, 014909 (2020)], we also include in the hydrodynamic evolution a temperature and baryon chemical potential dependent specific shear viscosity η/s .”

Comment #2:

2) Is the baryon number in the CF-transition conserved locally or globally? This might introduce additional correlations among the baryons, e.g. local conservation might lead to additional annihilation of baryons in the kinetic equation.

Authors: In our calculations, we have adopted the grand-canonical ensemble for particlization using the Cooper-Frye formula, which only allows the conservation of the average baryon number. The implementation of exact local baryon number conservation at the event-by-event level in the MUSIC package is still under development and is not yet available for the present study.

Changes made in response to comment #2: In the first paragraph of the section “Results and Discussion” on the left column of page 3, we have explained the modeling of partonic matter expansion and hadronization of the QGP by adding the sentences, “For the evolution of the QGP produced in these collisions, we use the viscous hydrodynamic package MUSIC [Phys. Rev. C 93(4), 044906 (2016); Phys. Rev. C 97(2), 024907 (2018); Nucl. Sci. Tech. 31(12), 122 (2020)] with the collision-geometry-based 3D initial conditions [Phys. Rev. C 102, 014909 (2020)] and a crossover equation of state NEOS-BQS [Phys. Rev. C 100(2), 024907 (2019)] with vanishing net strangeness density $n_s = 0$ and the net electric charge-to-baryon density ratio $n_Q = 0.4n_B$. Following Ref. [Phys. Rev. C 102(1), 014909 (2020)], we also include in the hydrodynamic evolution a temperature and baryon chemical potential dependent specific shear viscosity η/s . At hadronization of the QGP, both hadrons and light nuclei are produced in a grand-canonical ensemble using the statistical hadronization model in the grand-canonical ensemble approach [Phys. Rev. C 97(2), 024907 (2018)] with their phase-space distributions sampled on a constant energy density hypersurface according to the Cooper-Frye formula [Phys. Rev. D 10(1), 186–189 (1974)].”

Comment #3:

3) How have the statistical model results been restricted to midrapidity in Fig.3? I am asking because different methods can be used. E.g. one could integrate over all momenta and and take the ratios afterwards, however, using the (T,μ) extracted at midrapidity (often used but not correct) or one can use the differential thermal distribution incl. the longitudinal momentum dependence and restrict the integration to the region around midrapidity (rarely used, but more correct). Both approaches yield different results for the ratios due to the longitudinal flow. They should also clarify which flow profile was used.

Authors: In our calculation, the initial deuterons and tritons are sampled on the hydro hypersurface at hadronization using the Cooper-Frye formula. This sampling has been implemented in the hydrodynamic package MUSIC, and the effects of longitudinal flow are thus automatically taken into account. Like other particles, each deuteron or triton has a space-time position (x,y,z,t) and 4-momentum (E,px,py,pz) with its rapidity given by $y = \frac{1}{2} \ln\left(\frac{E+pz}{E-pz}\right)$. To obtain the deuteron and triton yields in mid-rapidity, we simply count their numbers in the midrapidity $|y| \leq 0.5$ region. With this method, we can also calculate the transverse momentum spectra of deuterons and tritons in our approach (see Fig. 2 in the reply). The statistical model curves in Fig. 3, which are obtained using (T,μ) extracted from particle yields at mid-rapidity, are taken from the STAR paper [Phys. Rev. Lett. 130, 202301 (2023)]

Comment #4:

4) The authors have omitted the data point at LHC. This is important because the cluster yields at LHC can be described by the statistical model and the RHIC data seems to be consistently lower than the LHC data. To make the point (as the authors do) that the RKE is needed to describe the data their approach should also be validated against the LHC data to allow for a firm conclusion.

Figure 1: Validation against the LHC data: Charged-particle multiplicity ($dN_{ch}/d\eta$) dependence of hadronic re-scattering effects on light nuclei yield ratios N_d/N_p (a), N_{3He}/N_p and N_{3H}/N_p (b). Theoretical results with and without including the hadronic dynamics are from Hydro+RKE (shaded bands) and the SHM [A. Andronic et al., “Decoding the phase structure of QCD via particle production at high energy”, Nature 561(7723), 321–330 (2018); M. Abdulhamid et al., (STAR Collaboration) “Beam Energy Dependence of Triton Production and Yield Ratio ($N_t \times N_p/N_d^2$) in Au+Au Collisions at RHIC”, Phys. Rev. Lett. 130, 202301 (2023)] (lines), respectively. The ALICE data are denoted by symbols with combined statistical and systematic uncertainties [J. Adam et al., (ALICE Collaboration) “Production of light nuclei and anti-nuclei in pp and Pb-Pb collisions at energies available at the CERN Large Hadron Collider”, Phys. Rev. C 93(2), 024917 (2016); S. Acharya et al., (ALICE Collaboration) “Light (anti)nuclei production in Pb-Pb collisions at $\sqrt{s_{NN}}=5.02$ TeV”, Phys. Rev. C 107(6), 064904 (2023)]. **[In response to Comment #4 of Reviewer 1, Comment #1 and #2 of Reviewer 2, and Comment #5 of Reviewer 3.]**

Authors: We thank the reviewer for this excellent suggestion and have validated our model calculation against the LHC data as shown in Fig. 1. These data are the latest ones from the ALICE Collaboration based on the measurement of the helium-3 and triton yields from Pb+Pb collisions at $\sqrt{s_{NN}} = 5.02$ TeV [Phys.Rev.C 107 (2023) 6, 064904], which was pub-

lished only a few months ago. The newly measured yield ratios N_{3He}/N_p and N_{3H}/N_p are systematically lower than the old data point in Pb+Pb collisions at $\sqrt{s_{NN}} = 2.76$ TeV and that from the SHM prediction. We emphasize that the previous statement that “the cluster yields at LHC can be described by the statistical model” is based on the old data at $\sqrt{s_{NN}} = 2.76$ TeV and is no longer supported by the new measurements. With the inclusion of hadronic re-scatterings, our kinetic approach (shaded bands) reproduces well the new data. Considering the large uncertainty of the old data point and the much higher statistics of the new data points, the comparison of our new theoretical results with the new experimental data further strengthens our conclusion that hadronic re-scatterings play an indispensable role in the little-bang nucleosynthesis.

Changes made in response to comment #4: On page 5, we have added a new section, namely, “Hadronic effects at the LHC energies”, together with a new figure (Fig. 4 in the revised manuscript) to discuss in detail the effects of hadronic re-scatterings on deuteron and triton production in Pb+Pb collisions at the LHC energies.

Comment #5:

5) *The results strongly depend on the total pion yield and how the pions are transported during the time evolution of the system. Because it might be that the pions are traveling in rho-mesons and do not participate in the reactions described in the paper or do the authors oversaturate they pion density because the pions are considered already free? The authors should therefore demonstrate that their final pion yield is comparable with the experimental results and should clarify what kind of pions are used.*

Authors: In our calculations, we have already taken into account the conversion among pions and rho mesons through the reactions $\rho \leftrightarrow \pi + \pi$ during the hadronic matter expansion. At hadronization, the rho to pion ratio N_ρ/N_π is about 0.2, and the rho mesons are gradually converted to pions as the density and temperature of the hadronic matter decrease during its fast expansion. We have also included other low-lying resonances like w , K^* , Δ , N^* , etc. Figure 2 shows the comparison of particle transverse momentum spectra between experimental data and the results from our model calculations. It is seen that the momentum spectrum of π^+ from our model is comparable to the experimental data up to 2 GeV. Our underestimate of the data above 2 GeV is likely due to the neglect of contributions from jet fragmentation in our study. Furthermore, we have tested the contribution of hadronic re-scatterings with rho meson as a catalyzer, e.g., $\rho + d \rightarrow \rho NN$, by assuming their cross sections to have similar values as those for pion catalyzed reactions, and we find their effects on the final deuteron and triton numbers to be negligible.

Comment #6:

6) *The connection to similar observations made from hadronic resonances (K^* ,...) should be discussed.*

Authors: We thank the reviewer for pointing out the connection of our results to observations made on the production of hadronic resonances, which also support the important

Figure 2: Comparison of particle momentum spectra in central Au+Au collisions at $\sqrt{s_{NN}} = 200$ GeV between data (symbols) and model calculation (lines). Experimental data points are taken from [J. Adam et al., (STAR Collaboration) “Beam energy dependence of (anti-)deuteron production in Au + Au collisions at the BNL Relativistic Heavy Ion Collider”, Phys. Rev. C99(6), 064905 (2019); M. Abdulhamid et al., (STAR Collaboration) “Beam Energy Dependence of Triton Production and Yield Ratio ($N_t \times N_p / N_d^2$) in Au+Au Collisions at RHIC”, Phys. Rev. Lett. 130, 202301 (2023)]. **[In response to Comment #5 of Reviewer 1.]**

effects of hadronic re-scatterings. Our study of hadronic effects on light nuclei production is complementary to similar studies on hadronic resonances, with both suggesting that the SHM is not directly applicable in describing the production of loosely-bound states or unstable particles that are emitted during the later stage of hadronic evolution.

Changes made in response to comment #6: In the right column of page 4, we have added the new paragraph, “Similar to deuteron and triton, short-lived hadronic resonances like $K^{*0}(890)$ and $\Lambda(1520)$ can frequently decay and reform via e.g., $K^{*0}(890) \leftrightarrow K^+ \pi^-$ and $\Lambda(1520) \leftrightarrow p K^-$ during the hadronic stage of relativistic heavy-ion collisions. Consequently, their numbers might change as the hadronic matter expands, and such hadronic effects have indeed been observed in experiments [Phys. Rev. C 91, 024609 (2015); Phys. Rev. C 99, 024905 (2019)].”

Comment #7:

7) *To summarize, the idea and the findings are interesting, but at the moment further information is needed before a conclusion can be drawn.*

Authors: We have demonstrated that our findings on the strong hadronic effects on light nuclei production in Au+Au collisions at RHIC energies are further supported by the latest ALICE data from Pb+Pb collisions at higher LHC energies. These latest data from both

RHIC and LHC allow us to draw a firm conclusion on the importance of hadronic dynamics in the little-bang nucleosynthesis.

REVIEWER 2

Introductory comment:

The manuscript presents a study of the production of light nuclei (deuteron, triton) in collisions of nuclei at large energies. It is a very relevant subject and the approach presented in the manuscript very good in principle and interesting, but I find that its strong claims are not backed by equally strong proofs.

Authors: We thank the reviewer's interest in our work and his/her careful reading of our manuscript as well as insightful, critical, and constructive feedback. We appreciate the opportunity that his/her comments have enabled us to prepare a greatly improved manuscript. In our revised manuscript, we have demonstrated that our conclusion on the strong hadronic effects on triton production in heavy ion collisions at relativistic energies is supported not only by the recent STAR data from Au+Au collisions at RHIC but also by the latest ALICE data from Pb+Pb collisions at the LHC. It can be seen from Fig. 1 in the reply that the four new data points on the yield ratios N_{3He}/N_p and N_{3H}/N_p measured by the ALICE collaboration in Pb+Pb collisions are significantly lower than the SHM predictions, which is similar to what one finds in Au+Au collisions at RHIC energies. With the inclusion of pion-catalyzed multi-nucleon scatterings, our model calculations can reproduce well the latest data on the triton and helium-3 yields. Adding the results from Pb+Pb collisions at the LHC energies to those presented in our original manuscript from Au+Au collisions at RHIC, we believe our study has provided a strong proof for our claim on the importance of hadronic re-scattering effects in the little-bang nucleosynthesis.

Comment #1:

The authors claim to explain the relative production of deuteron and triton nuclei to that of protons over a broad collision energy (per nucleon pair, $\sqrt{s_{NN}}$) range from about 8 to 200 GeV. The author's approach is to start from the predictions of the statistical hadronization model (SHM), which describes hadron production at the instant of the (sudden) chemical freeze-out (at a temperature of about 156 MeV, for the collision range explored here) and follow the dynamics of the system in its hadronic stage. The conclusion is that the predictions by SHM for the deuterons are not affected by the hadronic stage, while a strong reduction is observed for the tritons, in agreement with the data measured by the STAR collaboration.

Several similar studies have been published already, refs. [Phys. Lett. B 800, 135131 (2020); Phys. Lett. B 827, 136891 (2022); Phys. Rev. C 99(4), 044907 (2019)], focusing on the hadronic stage at LHC energies (2.76 TeV). In ref. [Phys. Lett. B 827, 136891 (2022)];, no effect is seen in the hadronic stage for both deuterons and tritons (while that study is in agreement with the ALICE data). The authors note this difference in the present manuscript, but give no clarification at all. The authors of the present manuscript have

shown in an earlier manuscript posted on arXiv (2106.12742, sharing some Figures with the present manuscript) a similarly good description of the data at the LHC (including also pp collisions).

And here lies my problem: I cannot see why there should be no effect at the LHC energies and a significant effect at lower energies, given that the hadronic system is at the energy of 200 GeV very similar to that of 2.76 TeV (with the exception of a small asymmetry in favor of matter compared to antimatter, but this cannot play a major role).

Authors: We appreciate the reviewer’s insightful comments and questions, which we address as follows.

Firstly, we would like to point out that the statement “no effect at the LHC energies” is based on the old data from Pb+Pb collisions at $\sqrt{s_{NN}} = 2.76$ TeV [Phys. Rev. C 93(2), 024917 (2016)] measured by the ALICE Collaboration, which is no longer true after the more precise measurements by the ALICE collaboration. According to the newly published data on the helium-3 and triton yields in Pb+Pb collisions at $\sqrt{s_{NN}} = 5.02$ TeV [Phys.Rev.C 107 (2023) 6, 064904], which is shown in Fig. 1 in the reply, the measured yield ratios N_{3He}/N_p and N_{3H}/N_p are systematically lower than the old data in Pb+Pb collisions at $\sqrt{s_{NN}} = 2.76$ TeV and also the predicted values from the SHM. Although the central value of N_{3He}/N_p in the single old data point at $\sqrt{s_{NN}} = 2.76$ TeV is much larger than those at $\sqrt{s_{NN}} = 5.02$ TeV, the two measurements at $\sqrt{s_{NN}} = 2.76$ TeV and $\sqrt{s_{NN}} = 5.02$ TeV are consistent within their uncertainties in view of the large uncertainty in the data at $\sqrt{s_{NN}} = 2.76$ TeV.

Secondly, we would like to emphasize that our kinetic approach with the inclusion of hadronic re-scatterings effects reproduces well the new data. Considering the large uncertainty in the old data point and the much higher statistics in the new data points, this new comparison strengthens our conclusion that hadronic re-scatterings play an indispensable role in the little-bang nucleosynthesis.

Thirdly, we want to mention that the data on the transverse momentum spectrum of helium-3 shown in arXiv:2106.12742 is for 0-20% centrality while our model calculations is for 0-10% centrality. As a result, the triton number in our model calculation is actually lower than that in the old data from Pb+Pb collisions at 2.76 TeV. Our results are, however, in good agreements with the new and more precise measurements at 5.02 TeV. We further note that the present work is based on the theoretical methods developed in our earlier manuscript (arXiv:2106.12742), which is, however, not under consideration for publication in any journals.

Lastly, the results obtained in earlier studies using the rate and Saha equations [Phys. Lett. B 800, 135131 (2020); Phys. Lett. B 827, 136891 (2022)] do not show any hadronic re-scatterings effects on triton production in relativistic heavy ion collisions. These model calculations are based on a simple expansion model for the hadronic fireball and invoke assumptions that may not hold near the kinetic freeze-out of the hadronic stage. Also, they fail to describe the new ALICE data (see Fig. 6 in the reply). By clarifying the importance of hadronic dynamics in the little-bang nucleosynthesis, we think our study will have a significant impact on the understanding of particle production in high-energy nuclear collisions.

Changes made in response to comment #1: On page 5, we have added a new section (“Hadronic effects at the LHC energies”) to discuss in detail the effects of hadronic re-scatterings on deuteron and triton production in Pb+Pb collisions at the LHC energies. Also included in this Section is an extension of the original paragraph on the difference between our approach and that using rate equations to “We notice that the reduced 3H number due to the hadronic effects in our results is absent in a recent study using the rate equation [Phys. Lett. B 827, 136891 (2022)] in a simple isentropic expansion model for the hadronic matter evolution. In this schematic approach, both kinetic thermal equilibrium and isentropic expansion are assumed for the hadronic matter when solving the rate equations for light nuclei dissociation and regeneration. These assumptions become questionable near the kinetic freeze-out when the system is driven out of chemical equilibrium. As a result, the absence of hadronic re-scattering effects in this approach leads to results that are in disagreement with the latest measurements on the triton and helium-3 numbers at RHIC and the LHC. ”

Comment #2:

In the similar approach for the deuterons published in ref. [Phys. Rev. C 99(4), 044907 (2019)] (which the authors cite properly), focused at the larger energy of the LHC where the pions dominate, it is pointed out the importance of the nucleons for the $d+X_j \rightarrow np$ reactions. Since at the lower energies explored in the present manuscript the proton/pion ratio is around 0.5 (at $\sqrt{s_{NN}}=8$), the channels involving nucleons ($X=n,p$) cannot be neglected. They are characterized by cross sections of similar magnitude as the $\pi+d$ reactions (see ref.[Phys. Rev. C 99(4), 044907 (2019)]) and are consequently expected to bring a significant contribution. This is another major shortcoming of the present manuscript.

Figure 3: Collision energy dependence of the yield ratio of nucleons to pions [L. Adamczyk et al., (STAR Collaboration) “Bulk Properties of the Medium Produced in Relativistic Heavy-Ion Collisions from the Beam Energy Scan Program”, Phys.Rev.C 96, 044904 (2017)]. **[In response to Comment #2 of Reviewer 2.]**

Authors: We appreciate the reviewer’s helpful comment. We have tested the effect of hadronic re-scatterings with the nucleon as a catalyzer and found its contribution to be negligible. Figure 3 displays the collision energy dependence of the yield ratio of nucleons to pions ($2N_p/3N_{\pi^+}$). For $\sqrt{s_{NN}}$ greater about 20 GeV, this ratio is about a few percent, which justifies the neglect of effects due to nucleon-induced hadronic re-scatterings. For collision energies at $7.7 \text{ GeV} \leq \sqrt{s_{NN}} \leq 20 \text{ GeV}$, this ratio becomes larger than 10% and reaches about 40% at 7.7 GeV. However, as shown in Fig. 4, the effect of enhanced scattering rates for deuteron and triton production turns out to be very small. This is because both dissociation and regeneration rates are enhanced by a similar amount, resulting in an almost cancellation of the two contributions. Consequently, the results presented in the original version of our manuscript remain unchanged and our conclusion also remains intact.

Figure 4: Time evolution of light nuclei yields in central Au+Au collisions at $\sqrt{s_{NN}} = 7.7$ GeV. Results with pion-catalyzed reactions are denoted by solid lines, while results with both pion and nucleon-catalyzed reactions are denoted by dashed lines. Experimental data points are taken from [J. Adam et al., (STAR Collaboration) “Beam energy dependence of (anti-)deuteron production in Au + Au collisions at the BNL Relativistic Heavy Ion Collider”, Phys. Rev. C99(6), 064905 (2019); M. Abdulhamid et al., (STAR Collaboration) “Beam Energy Dependence of Triton Production and Yield Ratio ($N_t \times N_p/N_d^2$) in Au+Au Collisions at RHIC”, Phys. Rev. Lett. 130, 202301 (2023)]. **[In response to Comment #2 of Reviewer 2.]**

Changes made in response to comment #2: In the left column of page 5, we have added the paragraph, “ It is worthwhile to mention that for Au+Au collisions at lower energies ($\sqrt{s_{NN}} \leq 20 \text{ GeV}$), the hadronic matter becomes more baryon rich, and hadronic re-scatterings with the nucleon as a catalyzer for light nuclei production is expected to play an increasingly important role. We find, however, that the resulting enhanced scattering rates for deuteron and triton production only have small effects on their final numbers even at the

low collision energy of 7.7 GeV. This is mainly because both dissociation and regeneration rates are enhanced at a similar level, resulting in an almost complete cancellation of their effects. ”

Comment #3:

Sure, the question remains why SHM explains the ^3He yield measured by ALICE at 2.76 TeV and not the measurements on tritons by STAR (ref. [Phys. Rev. Lett. 130, 202301 (2023)]), but the observation in the present manuscript of a large effect in the hadronic stage needs stronger proofs. In particular, the detailed balance for the reaction channels relevant for the tritons needs to be demonstrated (the authors of refs. [Phys. Lett. B 800, 135131 (2020); Phys. Lett. B 827, 136891 (2022); Phys. Rev. C 99(4), 044907 (2019)] put, rightly so, a great emphasis on this).

Authors: We thank the reviewer for his/her helpful comments. We have pointed out in the above that although the SHM can explain the ^3He yield (which has a very large uncertainty) at 2.76 TeV, it strongly overestimates the ^3He and ^3H yields newly measured at 5.02 TeV by the ALICE collaboration. Therefore, the SHM overpredicts the triton yield at all collision energies of $\sqrt{s_{NN}} = 7.7 - 5020$ GeV.

We agree with the reviewer that the detailed balance is crucial in the kinetic calculations. In a box calculation, we have demonstrated that the correct thermal limits of light nuclei yields are achieved. In Fig. 5, we show that the differential reaction rates of forward and backward scatterings are identical within the statistical uncertainties in our numerical calculations, suggesting that the principle of detailed balance is indeed preserved in our calculations.

Figure 5: Differential rates for the regeneration and dissociation of deuteron and triton in a box calculation. [In response to Comment #3 of Reviewer 2.]

Changes made in response to comment #3: On page 8, we have modified last sentence of the paragraph on the thermal limit in the box calculation to “It is seen that their final numbers are consistent with their expected thermal values at chemical equilibrium over five orders of magnitude, suggesting that the detailed balance is well preserved in our calculations.”

Comment #4:

Further aspects which I find weak in the present manuscript: The time scales needed for the equilibration in the hadronic stage shown in Fig. 2 are very long, implying a hadronic stage of about 25 fm/c. To me, this is very unrealistic, in particular since the measurements at the LHC (ALICE, arXiv:2211.04384) limit the hadronic phase to 10 fm/c and at lower energies the duration has to be smaller.

Figure 6: Longitudinal HBT radius (R_{long}) as a function of K_T . Symbols are obtained using the CRAB model together with the phase-space distribution of positively charged pions from our Hydro+RKE model. The solid line is from fitting the calculated R_{long} using $R_{long}^2 = \tau_f^2 \frac{T_{kin}}{m_T} \frac{K_2(m_T/T_{kin})}{K_1(m_T/T_{kin})}$ [M. Lisa, “Timescales in heavy ion collisions”, Acta Phys.Polon.B 47, 1847 (2016)] with $T_{kin} = 0.12$ GeV and $m_T = \sqrt{m_\pi^2 + K_T^2}$. **[In response to Comment #4 of Reviewer 2.]**

Authors: Thanks for pointing out this interesting and important issue. With some efforts, we have found that the relatively long hadronic phase does not contradict to that from the HBT measurements. We have verified this by using the CRAB model, which is a standard tool for evaluating the HBT correlations, to obtain the three-dimensional two-pion ($\pi^+ - \pi^+$) correlation function and extract the momentum dependence of the HBT radii (R_{out} , R_{side} , R_{long}). Figure 6 displays the momentum dependence of R_{long} , from which we have extracted a decoupling time $\tau_f \approx 8.4$ fm that is consistent with the measured value. This means the extracted value for τ_f from the HBT measurement is much smaller than the kinetic freezeout time at which particle interactions cease. There is therefore no contradiction between our results and the HBT measurements, and they are actually consistent with each other.

We would like to point out that, in the HBT measurements, one only probes the so-call homogeneity length in the emission source, which is known to be typically smaller than the geometric size of the expanding fireball. The lifetime is usually extracted from the approximate relation $R_{long}^2 = \tau_f^2 \frac{T_{kin}}{m_T} \frac{K_2(m_T/T_{kin})}{K_1(m_T/T_{kin})}$ [M. Lisa, Acta Phys.Polon.B 47, 1847 (2016)]. Hence, the lifetime extracted from HBT measurements could be much shorter than the kinetic freeze-out time of particles emitted from the expanding source.

We have also tried to calculate kinetic freeze-out time of emitted particles using transport

models like AMPT or hybrid models like EPOS, and they all give a similar long hadronic phase as in our model calculation.

Comment #5:

In Fig. 3, left plot, one observes an increase of the d/p ratio compared to the SHM prediction, it is not clear how this can happen, given the results shown in Fig. 2.

Authors: We again thank the reviewer for his/her comment. The larger d/p ratio compared to the SHM prediction as the collision energy increases is because the proton yield in our model calculation is slightly smaller than the SHM prediction at large collision energy.

Changes made in response to comment #5: In the right column of page 4, we have added the sentence, “The slightly larger N_d/N_p ratio from the Hydro+RKE than the SHM prediction at $\sqrt{s_{NN}} \geq 40$ GeV is due to the smaller proton number in our calculations.”

Comment #6:

Thermal limit in a box: 480 pions of each charge and 24 protons and neutrons are used, corresponding to the energy of 200 GeV (roughly at least), for one unit of rapidity. But the volume of 1 unit of rapidity at chemical freeze-out (which describes the abundancies quoted above) is about 2000 fm³, while the authors use here (strangely and without any justification) 8000 fm³!

Authors: Thank you very much for pointing out this issue. We are sorry that our explanation for the box calculation was not clear enough and has thus led to some confusions. In our box calculation, the box volume has no effects on the results and thus the validation of our method, because of the use of a periodic boundary condition in our calculations. In our setup for the box calculation, 24 protons in 8000 fm³ corresponds to an almost vanishing chemical potential $\mu_N \approx 0.007$ GeV, similar to the values extracted from heavy ion collisions at top RHIC energies and the LHC energies. The p/π^+ ratio in our calculation is set to be 0.05 as suggested from experimental measurements.

Changes made in response to comment #6: On page 8, we have modified the paragraph on the thermal limit in box calculations to “To validate the above stochastic method, we consider deuteron and triton production in a $(20 \text{ fm})^3$ box with periodic boundary conditions. Initially, 24 protons, 24 neutrons, and 480 pions for each of its three charge states are uniformly distributed in the box, corresponding to an almost vanishing chemical potential $\mu_B \approx 0.007$ GeV at top RHIC and the LHC energies. Their initial momentum distributions are taken to have the thermal Boltzmann form with temperature $T = 155$ MeV. The right panel of Fig. 6 shows the time evolution of the p , d , and ^3H numbers. It is seen that their final numbers are consistent with their expected thermal values at chemical equilibrium over five orders of magnitude, suggesting that the detailed balance is well preserved in our calculations. ”

Comment #7:

No explanation is given on the uncertainty bands shown in Fig. 2 and 3.

Authors: Thank you very much for the comment. The uncertainty bands in our kinetic calculations are purely statistical. Since these light nuclei are rarely produced in high-energy nuclear collisions, our statistics are limited by the available computational power.

Changes made in response to comment #7: On page 3, we have modified the last sentence in the caption of Fig. 2 to explain the uncertainty bands, namely, “Experimental data with combined statistical and systematic uncertainties from Refs. [Phys. Rev. Lett. 130, 202301 (2023); Phys. Rev. C 99(6), 064905 (2019)] are denoted by filled symbols, while theoretical results with statistical uncertainties are shown by shaded bands.”.

REVIEWER 3

Introductory comment:

The article titled “Unveiling the dynamics of little-bang nucleosynthesis” investigates the production of light nuclei from the quark-gluon plasma (QGP) through the statistical hadronization model. Specifically, the article investigates the mechanism of hadronic re-scattering and its effect on equilibrium light nuclei yields. The research direction is timely, motivated by the recent measurement by the STAR collaboration reporting the triton production over a wide range in collision energies. The precise production mechanism of light nuclei in heavy-ion collisions has been a topic of great debate and has therefore garnered significant experimental and theoretical interest in recent years. The contribution here potentially makes an important stride towards understanding the light nuclei production mechanism. This article addresses topics of interest in high energy nuclear physics with implications for the interpretation of light nuclei production in heavy-ion collision experiments.

*The studies presented in this article appear to be high quality with appropriate data and methods provided to clearly communicate the claims in the paper. However, based on the questions/comments listed below I find that the level of support for the primary conclusions could be improved through additional checks and comparisons with existing experimental data. The results are significant and make important progress in understanding the production mechanisms leading to the final yield of light nuclei in high-energy heavy-ion collisions a topic of intense interest. The methods employed and the presentation of their results are clear with a few exceptions – see the questions/comments below related to the data presentation and the uncertainty on the theoretical curves. Overall, I find the article well written and organized. My main concern with the article and the conclusions relates to the ability to describe the LHC data at higher energies (2.76 TeV) and the multiplicity dependence of the $N_{3H} * N_p/N_d^2$ data. The article may be suitable for publication if these points can be adequately addressed.*

Authors: We are delighted by the reviewer’s appreciation of our work and also his/her positive and insightful comments. Accordingly, we have improved our manuscript to fully address the reviewer’s comments and suggestions. Especially, we have provided further evidence of hadronic effects on light nuclei production in Pb+Pb collisions at higher LHC

energies.

Comment #1:

1. A few key statements come across too strong in pg2 rhs middle paragraph: "This effect has indeed been observed in recent experimental measurements at the Relativistic Heavy Ion Collider (RHIC)" and in the conclusions: "These distinct hadronic effects on deuteron and triton production have been confirmed in recent measurements by the STAR collaboration at RHIC."

I find these statements too strong, since coalescence and transport models are also in good agreement. What this article does not seem to address is any way of distinguishing these models as the true mechanism (or otherwise unifying their interpretation in some way, see my note below). Some comment on this consistency with respect to the coalescence model is given on page 4 lhs middle.

More neutral statement like the one just before "Results and Discussion" seem more appropriate - i.e. that the existing STAR results are in excellent agreement with this prediction: "However, the initial triton yield from hadronization of the QGP is reduced by about a factor of 1.8 during the hadronic matter expansion, leading to a result that is in excellent agreement with the latest measurements by the STAR Collaboration [Phys. Rev. Lett. 130, 202301 (2023)]."

In my opinion these statements would leave a reader without additional domain specific knowledge with the impression that the exact mechanism of light nuclei is established with no debate. This seems to be over stating the situation since, for instance the experimental data for light nuclei yields can be equally well explained by coalescence or hadronic transport models. If the authors mean by page 4 lhs middle that this finding unites or unifies the interpretation of these models then it should be made a more specific claim/statement of the article.

Authors: We thank the reviewer for his/her helpful comments and suggestions. In the revised manuscript, we have accordingly modified our statements to more neutral ones, which, however, do not weaken or affect the significance of our work.

Changes made in response to comment #1:In the right column of page 2, we have removed the statement "This effect has indeed been observed in recent experimental measurements at the Relativistic Heavy Ion Collider (RHIC)". In addition, we have rewritten in the conclusion the statement from "These distinct hadronic effects on deuteron and triton production have been confirmed in recent measurements by the STAR collaboration at RHIC." to "These distinct hadronic effects on deuteron and triton production are in excellent agreement with recent measurements by the STAR collaboration at RHIC and are further supported by the latest measurements by the ALICE collaboration at the LHC."

Comment #2:

2. STAR data: - What exactly is plotted in Fig. 2 with respect to the STAR datapoint from [Phys. Rev. Lett. 130, 202301 (2023)] - as I could not find points in the STAR manuscript

corresponding exactly to these values? Are they integrated yields (over pT), and if so is there some extrapolation uncertainty? Do the data point's error bars show combined statistical and systematic uncertainties, if so that should be noted somewhere in the caption of Fig. 2 (and for other figures).

Authors: The STAR data points plotted on Fig. 2 are the integrated yields over the transverse momentum. The data shown in the STAR paper [Phys. Rev. Lett. 130, 202301 (2023)] can be found on <https://www.hepdata.net/record/ins2152917>. The triton yields can also be found in this web page, although only the yield ratios are displayed in the STAR paper. The deuteron yields are taken from <https://www.hepdata.net/record/ins1727273> [Phys. Rev. C 99, 064905 (2019)]. Specifically, the experimental values for the deuteron and triton yields (dN/dy) in 0-10% centrality at $\sqrt{s_{NN}} = 200$ GeV are $0.0731828 \pm 9.99382 \times 10^{-5}(\text{stat.}) \pm 0.00528862(\text{sys.})$ and $0.0001175 \pm 0.0000028(\text{stat.}) \pm 0.000016(\text{sys.})$, respectively. In plots shown in our paper, we have combined statistical and systematic uncertainties together for experimental data points.

Changes made in response to comment #2: In the right column of page 3, we have added the sentence, “ which are denoted by filled symbols together with their combined statistical and systematic uncertainties.”. We have also modified the last sentence in the caption of Fig. 2 to explain the uncertainty bands, namely, “Experimental data with combined statistical and systematic uncertainties from Refs. [Phys. Rev. Lett. 130, 202301 (2023); Phys. Rev. C 99(6), 064905 (2019)] are denoted by filled symbols, while theoretical results with statistical uncertainties are shown by shaded bands.”.

Comment #3:

3. Multiplicity dependence Fig. 3 of the STAR triton paper [Phys. Rev. Lett. 130, 202301 (2023)] shows that the thermal model has a totally incorrect dependence on multiplicity ($dN_{ch}/d\eta$) compared to data. The article discusses the impact of the hadronic re-scattering with comparisons in the 0-10% most central collisions where the authors note the excellent agreement. But what about other centralities (lower particle multiplicities) and generally if the effect leads to an accurate multiplicity dependence? If not is there a justification for this?

Authors: Our model calculation can in principle be extended to any centrality. However, exact baryon number conservation has not been implemented in the present MUSIC package for hadronization/particization using the Cooper-Frye prescription. Calculation of light nuclei production in peripheral collisions using present MUSIC package will thus introduce systematic uncertainties in the results. Also, fine tuning of hydro parameters is required for describing the bulk evolution of quark-gluon plasma in peripheral collisions e.g., centrality larger than 40%. For central collisions, the canonical effects due to charge conservation are, on the other hand, known to be small, and the present MUSIC package can be used as it is. We thus restrict our calculation to the most central collisions. We have checked that the yield ratio $N_t \times N_p / N_d^2$ in our calculation is consistent with the experimental measurements at e.g., 30-40% centrality. Extending the present Hydro+RKE approach to the most peripheral collisions is a direction we plan to pursue in the near future.

Changes made in response to comment #3: In the right column of page 5, we have added a new paragraph before the conclusion section, “In the above calculations, we have only considered central collisions where canonical effects due to charge conservation at hadronization of the QGP are small [Phys. Lett. B 785, 171–174 (2018)]. Extending our study to peripheral collisions by including the canonical effects at hadronization is of great interest for future studies.”

Comment #4:

4. Source of uncertainty bands and some discussion In Fig. 2 and 3 the results are shown as shaded bands but there is no discussion of the source of spread (uncertainty) in the calculation. Since the SHM curve appears to be a line (not a band) it is presumably from the statistical error on the stochastic integration? It should be commented as to what contributes and if any other model uncertainties are included in this band. This uncertainty becomes especially interesting and important with respect to the discussion of the QGP critical point search where the uncertainty on the 'almost flat' prediction is of the same order as the uncertainty on the data itself.

Authors: Thank you very much for these comments. The uncertainty bands in Fig. 2 of our paper are entirely due to limited statistics in our kinetic calculation. Because light nuclei are rarely produced in high-energy nuclear collisions, the width of the uncertainty band thus depends on the number of events we have generated from the computational power available to us. In contrast, the SHM curves shown in Fig. 3 in our paper are taken from the STAR paper (Phys. Rev. Lett. 130, 202301 (2023)), and no uncertainty bands are displayed.

Changes made in response to comment #4: On page 3, we modified the last sentence in the caption of Fig. 2 to explain the uncertainty bands: “while theoretical results with statistical uncertainties are shown by shaded bands.” .

Comment #5:

5. Consistency with the LHC measurements from the STAR triton paper [Phys. Rev. Lett. 130, 202301 (2023)] : "Quantitatively, the thermal model describes the Nd/Np ratios well, but it systematically overestimates the Nt/Np ratios except for the results from central Pb + Pb collisions at $\sqrt{s_{NN}} = 2.76$ TeV" Based on this discussion and Fig. 3 in this article it seems that these hadronic re-scattering effects have a nearly beam-energy independent effect on $N(3H) / Np$ - reducing the SHM yield uniformly by a factor of 1.8. In fact the article says: "As can be inferred from Fig. 3, similar hadronic effects on triton production are expected at both lower and higher beam energies than those covered in the RHIC Beam Energy Scan program." Some comment should be made about the LHC measurement at 2.76 TeV. If the existing thermal model results are already in good agreement with the measurement at that energy it would imply that this dynamical effect would lead to an underestimation of the yield measured at 2.76 TeV. The article discusses the energy dependence, so presumably the comparison can be extended to these higher energy collisions? I would be interested in

seeing such a comparison. Regardless of whether or not it is added to the Fig 3. it could be commented on in the article.

Authors: We thank the reviewer for this excellent suggestion. We have extended our study from RHIC energies to higher LHC energies, and the results are displayed in Fig. 1. The statement that the existing thermal model results are in good agreement with the measurement at the LHC energies is based on the old RUN I data from Pb+Pb collisions at $\sqrt{s_{NN}} = 2.76$ TeV. This data point has a rather large uncertainty due to its limited statistics. In contrast, the newly measured triton and helium-3 yields at $\sqrt{s_{NN}} = 5.02$ TeV in the LHC RUN 2 experiment, which are much more precise, are systematically lower than the SHM prediction. Similar to the situation at RHIC energies, such a discrepancy can be resolved after taking into account the effects due to hadronic re-scatterings. Considering the much higher statistics in the new ALICE data, it becomes clear that the strong hadronic effects on triton production found in our model calculation are in excellent agreement with the latest measurements at both RHIC and LHC energies ($\sqrt{s_{NN}} = 7.7 - 5020$ GeV).

Changes made in response to comment #5: On page 5, we have added a new section (“Hadronic effects at the LHC energies”) to demonstrate in detail that similar hadronic re-scattering effects on deuteron and triton production are also found at the LHC energies.

Comment #6:

6. Clarity about the key difference of this approach. pg.2 rhs: "the post-hadronization dynamics has only small effects on their final abundance. This result is supported by studies using the Saha equation [Phys. Lett. B 800, 135131 (2020)] and the rate equation [Phys. Lett. B 827, 136891 (2022)] as well as the more microscopic transport approach [Phys. Rev. C 99(4), 044907 (2019)]."

page 4 lhs: "However, the decreased 3H yield due to the hadronic effects in our results is almost absent in a recent study using the rate equation [Phys. Lett. B 827, 136891 (2022)] in a simple isentropic expansion model for the hadronic matter evolution." similarly in the STAR paper (references refer therein): "Dynamical model calculations with hadronic rescatterings implemented using both the saha [Phys. Rev. C 50(4), 1796–1806 (1994)] and rate equations [Phys. Lett. B 827, 136891 (2022)] show that the deuteron, triton, and helium-3 yields remain unchanged during hadronic expansion." and the article comments on this in p2 rhs top paragraph. However, the reader is left wondering how two completely different results can be found for the effect of hadronic re-scattering (those quoted above) and that found in this article? It would be helpful to comment on the potential source of this discrepancy.

Authors: We thank the reviewer for this helpful comment. In our calculation, all dynamical effects like fireball expansion, radial flow, resonances decay, and the non-equilibrium nature near the kinetic freeze-out are fully taken into account in a self-consistent way. In contrast, in the study using the Saha or rate equations [Phys. Lett. B 827, 136891 (2022)], the radial flow is neglected and the fireball is assumed to expand isentropically. More importantly, the kinetic thermal equilibrium is imposed when solving the rate equations during the hadronic matter expansion. Figure 7 shows the temperature dependence of light nuclei yields. Results

Figure 7: Temperature dependence of light nuclei yields. Results using rate equations are denoted by lines, and they are from private communication with authors of [T. Neidig et al., “Towards solving the puzzle of high temperature light (anti)-nuclei production in ultra-relativistic heavy ion collisions”, Phys. Lett. B 827, 136891 (2022)]. Data on deuteron and triton (shaded bands) are taken from the latest measurements by the ALICE collaboration [S. Acharya et al., (ALICE Collaboration) “Light (anti)nuclei production in Pb-Pb collisions at $\sqrt{s_{NN}} = 5.02$ TeV”, Phys. Rev. C 107(6), 064904 (2023)].

using rate equations are denoted by lines ([Phys. Lett. B 827, 136891 (2022)]). Data on the deuteron and triton yields (shaded bands) are taken from the latest ALICE collaboration [Phys. Rev. C 107(6), 064904 (2023)]. It can be seen that the deuteron and triton numbers start to rise at $T < 100$ MeV instead of approaching to constant values. The light nuclei do not freeze out even at $T < 70$ MeV, suggesting that the calculation in Ref. [Phys. Lett. B 827, 136891 (2022)] is questionable when the system becomes out of thermal equilibrium.

It is also seen from Fig. 7 that the helium-3 yield from this simplified approach is much larger than the new ALICE data. In contrast, the deuteron and triton yields in our study approach to constant values near the kinetic freeze-out, and their final values agree with the latest ALICE data.

Changes made in response to comment #6: We have added in the right column of page 5 a paragraph to address the difference between our approach and that using rate equations, “We notice that the reduced 3H number due to the hadronic effects in our results is absent in a recent study using the rate equation [Phys. Lett. B 827, 136891 (2022)] in a simple isentropic expansion model for the hadronic matter evolution. In this schematic approach, both kinetic thermal equilibrium and isentropic expansion are assumed for the hadronic matter when solving the rate equations for light nuclei dissociation and regeneration. These assumptions become questionable near the kinetic freeze-out when the system is driven out of chemical equilibrium. As a result, the absence of hadronic re-scattering effects in this approach leads to results that are in disagreement with the latest measurements on the

triton and helium-3 numbers at RHIC and the LHC.”

Comment #7:

Finally I have a few specific comments on wording and/or grammar that could improve the flow and readability of the manuscript: “and etc.” - maybe provide one more example and drop the “and etc.” either way it should be “etc.” (without and)

“is a smooth crossover located at $T = 156.5$ ” - located only makes sense if you also say “in the phase diagram”, consider “... characterized by $T = \dots$ ”

*“Analyses of experimental data based on this statistical hadronization model” - “this”- $\dot{\jmath}$ “the”
“rescattering” vs. “re-scattering” be consistent*

*In several places “the” is used when it should be omitted and vice versa, e.g.: “The small hadronic effects on the deuteron abundance is due to similar deuteron dissociation and re-generation rates during the hadronic evolution” - $\dot{\jmath}$ “... hadronic evolution” (remove the)
“The equilibrated triton number (N_{3H}) can then be related to the proton number (N_p) and deuteron number (N_d)” - $\dot{\jmath}$ missing “the” - $\dot{\jmath}$ “.. and the deuteron number ... “*

“the inclusion of hadronic effects, the triton yields at all collision energies” - $\dot{\jmath}$ “..., triton yields at all collision energies”

“its greater sensitivity to the hadronic dynamics than nucleons and deuterons” - $\dot{\jmath}$ “its greater sensitivity to hadronic dynamics than nucleons and deuterons” (remove the)

“As to the triton yield” - $\dot{\jmath}$ “For the triton yield”

Authors: Thanks for a careful reading of our manuscript and the comments on some of the wordings and grammars. We have corrected these blunders and also tried our best to improve the manuscript by making additional changes to the manuscript. These changes will not influence the content and conclusion of our paper.

In summary, we thank the reviewers again for their valuable and helpful comments and suggestions. We have improved the manuscript accordingly and hope that they find their concerns about our study have all been properly addressed.

REVIEWER COMMENTS

Reviewer #1 (Remarks to the Author):

I would like to thank the authors for the careful reply to my previous comments and for the related modifications to the manuscript. From my point of view, the authors have addressed all my questions in a very clear way and have improved the manuscript to allow for publication.

In line with my previous evaluation, I am now even more convinced that this work is important and significant.

Therefore, I recommend to accept the current version of the paper for publication in Nature.

Best regards,

Marcus Bleicher

Reviewer #2 (Remarks to the Author):

I thank the authors for having thoroughly addressed my comments. I am satisfied by all the answers, but I am unconvinced about the answer my Comment #4, on the duration of the hadronic stage. I am of course aware of the caveats of the lifetime extraction from HBT measurements and I should have been more precise in my comments: I was referring to the estimates of the hadronic phase duration based on short-lived hadronic resonances, K^* , ρ and Λ^* , see Fig. 42 of arXiv:2211.04384 and associated discussion. Also there is to see that the calculations with EPOS3+UrQMD model indicate durations of up to 10 fm/c. I don't understand therefore the claim of the authors that EPOS gives a long hadronic phase. I think this remains rather problematic, since a hadronic lifetime of 7-8 fm/c roughly expected for 200 GeV based on the LHC results would compromise the success of the model reported in the manuscript. Since this is crucial, I would like to ask the authors to still address the issue.

Reviewer #3 (Remarks to the Author):

The authors have provided a significantly revised manuscript for “Unveiling the dynamics of little-bang nucleosynthesis” taking into account comments from three reviewers. Overall, I find the paper now includes more accurate and reliable conclusions. A primary concern expressed by all three reviewers was with regards to the (dis)agreement of the current approach with measurements at LHC energies. The updated manuscript includes a new section “Hadronic effects at the LHC energies” which addresses this concern by demonstrating that the proposed approach provides simultaneous description of the lower (RHIC) energy measurements, as well as the higher (LHC) energy measurements in light of the new, more precise results from ALICE at 502 TeV.

My other main concern was with respect to the difference of this approach compared to that in references e.g. [Phys. Lett. B 827, 136891 (2022)]. The paragraph added in the right column of page 5 has helped to clarify the key differences and therefore illuminate the essential physics.

The updates have addressed my concerns and present a coherent, interesting, well written analysis. I am now happy to recommend the article for publication in Nature Communications.

Reply (NCOMMS-23-24526A)

We thank the reviewers for their constructive comments and kind recommendations on our manuscript. In the revised manuscript, we have addressed the question raised by the second reviewer, which we believe has further enhanced the quality of our study. Detailed responses and revisions corresponding to each comment are provided below.

REVIEWER 1

Comment #:

I would like to thank the authors for the careful reply to my previous comments and for the related modifications to the manuscript. From my point of view, the authors have addressed all my questions in a very clear way and have improved the manuscript to allow for publication. In line with my previous evaluation, I am now even more convinced that this work is important and significant. Therefore, I recommend to accept the current version of the paper for publication in Nature.

Authors: We are delighted by the reviewer’s appreciation of our work and positive recommendation on our paper.

REVIEWER 2

Comment #:

I thank the authors for having thoroughly addressed my comments. I am satisfied by all the answers, but I am unconvinced about the answer my Comment #4, on the duration of the hadronic stage. I am of course aware of the caveats of the lifetime extraction from HBT measurements and I should have been more precise in my comments: I was referring to the estimates of the hadronic phase duration based on short-lived hadronic resonances K^ , ρ and Λ^* , see Fig. 42 of arXiv:2211.04384 and associated discussion. Also there is to see that the calculations with EPOS3+UrQMD model indicate durations of up to 10 fm/c. I don’t understand therefore the claim of the authors that EPOS gives a long hadronic phase. I think this remains rather problematic, since a hadronic lifetime of 7-8 fm/c roughly expected for 200 GeV based on the LHC*

Figure 1: (a) Time distribution of kinetic freeze-out particles in central Au+Au collisions at 200 GeV from our Hydro+RKE model calculation (solid line) and the EPOS4+UrQMD calculation (dashed line). (b) Comparison of hadronic matter lifetime between model calculations and experimental results extracted from hadronic resonances’ relative abundance (data taken from Fig. 42 of [ALICE Collaboration, “The ALICE experiment – A journey through QCD”, arXiv:2211.04384]).

results would compromise the success of the model reported in the manuscript. Since this is crucial, I would like to ask the authors to still address the issue.

Authors: We are grateful to the reviewer’s insightful comments and questions. In the following, we explain in detail why the seemingly long hadronic duration time in Fig. 2 of our paper does not contradict the short lifetime of the hadronic phase estimated based on measured yields of short-lived hadronic resonances and the calculations with the EPOS3+UrQMD model.

We first show in Fig. 1(a) the kinetic freeze-out time distribution of hadronic particles in central Au+Au collisions at $\sqrt{s_{NN}} = 200$ GeV from our Hydro+RKE model (solid line) and the EPOS4+UrQMD model (<https://klaus.pages.in2p3.fr/epos4/>) (dashed line). It shows that, hadronic particles in both models are emitted over a very broad time interval with similar average freeze-out time of approximately 21 fm/c and 19.3 fm/c in the Hydro+RKE and the EPOS4+UrQMD model, respectively. This similarity in the kinetic freeze-out time distribution and average kinetic freeze-out lifetime in these two models is not surprising because both models have similar partonic and hadronic dynamics.

We then show in Fig. 1(b) the hadronic matter lifetime as a function of $dN_{ch}/d\eta$ from measured short-lived hadronic resonances (Λ^* , ρ^0 , and K^{*0}) (solid squares), the EPOS3+UrQMD model (solid line), and our Hydro+RKE mode (solid star). These results are obtained by using the exponential decay formula $r_{\text{kin}} = r_{\text{chem}} \times e^{-(\tau_{\text{kin}} - \tau_{\text{chem}})/\tau_{\text{res}}}$ of the ALICE Collaboration [arXiv:2211.04384] from measured Λ^*/Λ , ρ^0/π and K^{*0}/K yield ratios, and the formula $\tau_{\text{hadronic duration}} = \tau_{\text{kinetic freezeout}} - \tau_{\text{hadronization+direct decay}}$ using the average hadronization and kinetic freeze-out times [Phys.Rev.C 93, 014911 (2016)] for the EPOS3+UrQMD and Hydro+RKE models. It is seen that the about of 9 fm/c of hadronic matter lifetime from our Hydro+RKE model is consistent with those obtained from both the EPOS3+UrQMD model and measured Λ^*/Λ yield ratio. Although this 9 fm/c hadronic phase lifetime is longer than those estimated from the ρ^0/π and K^{*0}/K yield ratios, it is important to note that the exponential decay formula used in the estimation of hadronic phase lifetime gives only a lower limit of hadronic duration time, as it neglects the effect of regeneration and assumes sharp hadronization and kinetic freeze-out times for all particles. The long hadronic duration time in our Hydro+RKE model is therefore a result of the broad hadronization and kinetic freeze-out times. The difference between the average hadronization and kinetic freeze-out times in our model is, however, about a factor of two shorter, similar to the result from the EPOS3+UrQMD model and from the measured yield ratios of resonances to their stable ground state hadrons. As demonstrated in our previous reply, the HBT radius from the two-pion correlation using freeze-out pions in our model is consistent with that extracted from experimental data.

Changes made in response to comment #1: To avoid the impression by readers that the hadronic matter lifetime in our Hydro-RKE model is much longer than those from the EPOS3+UrQMD model and from the measured yield ratios of resonances to their stable ground state hadrons, we have rewritten the paragraph above the section on “Collision energy dependence” to “Similar to deuteron and triton, short-lived hadronic resonances like $K^{*0}(890)$, ρ , and $\Lambda(1520)$ can frequently decay and reform via e.g. $K^{*0}(890) \leftrightarrow K + \pi$, $\rho \leftrightarrow \pi\pi$, and $\Lambda(1520) \leftrightarrow pK$. Consequently, their numbers might change as the hadronic matter expands, and such hadronic effects have indeed been observed in experiments [Phys. Rev. C 91, 024609 (2015); Phys. Rev. C 99, 024905 (2019)]. Based on the exponential decay law for hadronic resonances that neglects their regenerations during the hadronic evolution, the extracted hadronic matter lifetime in relativistic heavy-ion collisions is less than 10 fm/c [arXiv:2211.04384]. A similar short hadronic matter lifetime is found in Ref. [Phys.Rev.C 93, 014911 (2016)] using the EPOS+UrQMD model by taking the hadronic matter lifetime to be the difference between the average kinetic freeze-out time and the sum of the average hadronization time and the resonance lifetime. Using the latter definition in our Hydro+RKE model also gives a short hadronic matter lifetime of about 9 fm/c. Because of the broad kinetic freeze-out time distributions in Hydro+RKE, particularly for deuteron and triton, to properly take into account the hadronic recattering effects on their final yields requires following the time evolution of the hadronic matter to a much longer time than their mean kinetic freeze-out time as shown in Fig. 2.”

Comment:

The authors have provided a significantly revised manuscript for “Unveiling the dynamics of little-bang nucleosynthesis” taking into account comments from three reviewers. Overall, I find the paper now includes more accurate and reliable conclusions. A primary concern expressed by all three reviewers was with regards to the (dis)agreement of the current approach with measurements at LHC energies. The updated manuscript includes a new section “Hadronic effects at the LHC energies” which addresses this concern by demonstrating that the proposed approach provides simultaneous description of the lower (RHIC) energy measurements, as well as the higher (LHC) energy measurements in light of the new, more precise results from ALICE at 5.02 TeV. My other main concern was with respect to the difference of this approach compared to that in references e.g. [Phys. Lett. B 827, 136891 (2022)]. The paragraph added in the right column of page 5 has helped to clarify the key differences and therefore illuminate the essential physics. The updates have addressed my concerns and present a coherent, interesting, well written analysis. I am now happy to recommend the article for publication in Nature Communications.

Authors: We are delighted by the reviewer’s appreciation of our work and positive recommendation on our paper.

REVIEWERS' COMMENTS

Reviewer #2 (Remarks to the Author):

I thank the authors for thoroughly have addressed my comment. I'm happy to recommend the publication in Nature Communications.

Reply (NCOMMS-23-24526B)

REVIEWER 2

Comment:

I thank the authors for thoroughly have addressed my comment. I'm happy to recommend the publication in Nature Communications.

Authors: We are grateful to the reviewer for his/her positive recommendation on our paper.